# Data Continuity Matters: Improving Sequence Modeling with Lipschitz Regularizer

**Eric Qu**[*]
Duke Kunshan University
Kunshan, Jiangsu 215316, China
`zhonghang.qu@duke.edu`

**Xufang Luo**[†]**& Dongsheng Li**
Microsoft Research Asia
Shanghai 200232, China
`{xufluo, dongsheng.li}@microsoft.com`

## Abstract

Sequence modeling is a core problem in machine learning, and various neural networks have been designed to process different types of sequence data. However, few attempts have been made to understand the inherent data property of sequence data, neglecting the critical factor that may significantly affect the performance of sequence modeling. In this paper, we theoretically and empirically analyze a generic property of sequence data, i.e., *continuity*, and connect this property with the performance of deep models. First, we empirically observe that different kinds of models for sequence modeling prefer data with different continuity. Then, we theoretically analyze the continuity preference of different models in both time and frequency domains. To further utilize continuity to improve sequence modeling, we propose a simple yet effective *Lipschitz Regularizer*, that can flexibly adjust data continuity according to model preferences, and bring very little extra computational cost. Extensive experiments on various tasks demonstrate that altering data continuity via Lipschitz Regularizer can largely improve the performance of many deep models for sequence modeling.[1]

## 1 Introduction

Sequence modeling is a central problem in many machine learning tasks, ranging from natural language processing (Kenton & Toutanova, 2019) to time-series forecasting (Li et al., 2019). Although simple deep models, like MLPs, can be used for this problem, various model architectures have been specially designed to process different types of real-world sequence data, achieving vastly superior performance to simple models. For instance, the vanilla Transformer shows great power in natural language processing (Wolf et al., 2020), and its variant Informer (Zhou et al., 2021) is more efficient in time-series forecasting tasks. And a recent work Structured State Space sequence model (S4) (Gu et al., 2021) reaches SoTA in handling data with long-range dependencies. However, few attempts have been made to understand the inherent property of sequence data in various tasks, neglecting the critical factor which could largely influence the performance of different types of deep models. Such investigations can help us to understand the question that what kind of deep model is suitable for specific tasks, and is essential for improving deep sequence modeling.

In this paper, we study a generic property of sequence data, i.e., *continuity*, and investigate how this property connects with the performance of different deep models. Naturally, all sequence data can be treated as discrete samples from an underlying continuous function with time as the hidden axis. Based on this view, we apply continuity to describe the smoothness of the underlying function, and further quantify it with Lipschitz continuity. Then, it can be noticed that different data types have different continuity. For instance, time-series or audio data are more continuous than language sequences, since they are sampled from physical continuous signals evolved through time.

Furthermore, we empirically observe that *different deep models prefer data with different continuity*. We design a sequence-to-sequence task to show this phenomenon. Specifically, we generate

---

[*]Work done during an internship in Microsoft Research Asia.
[†]Corresponding author.
[1]Code is available at https://EricQu.site/LipReg/

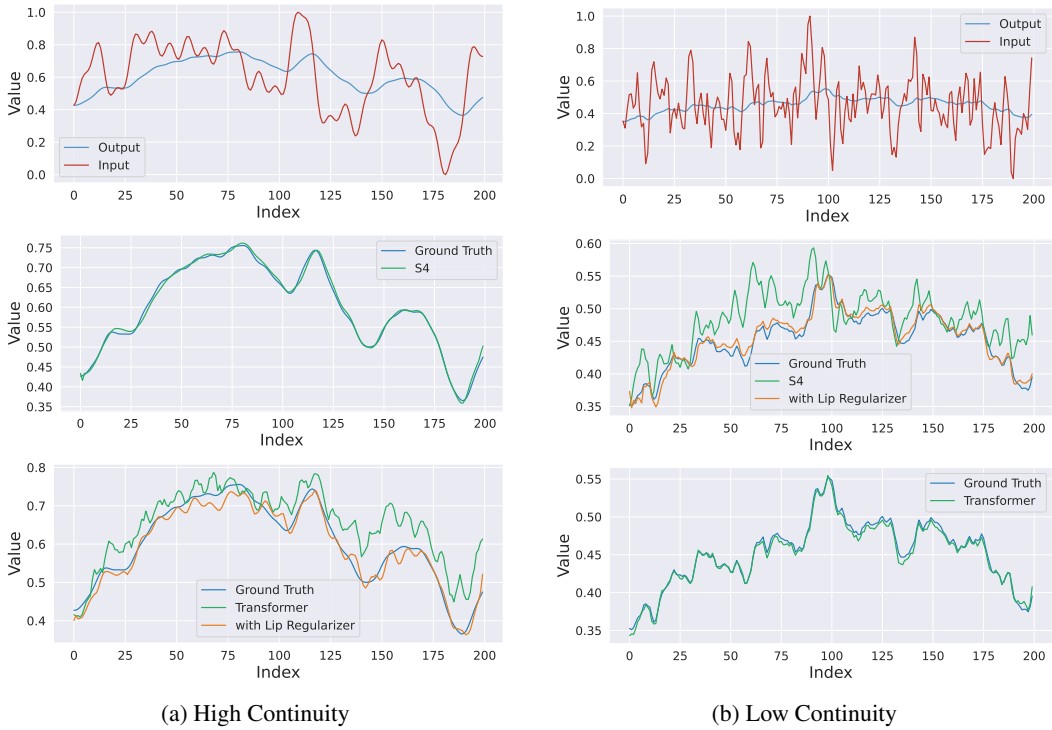

(a) High Continuity        (b) Low Continuity

Figure 1: A Sequence-to-sequence task to show that different deep models prefer data with different continuity. The first row shows input and output sequences. We generate input sequences with different continuities (left column: high continuity; right column: low continuity) and learn a mapping function using different models (second row: S4; third row: Transformer). We can see that S4 prefers more continuous sequences, while Transformer prefers more discrete sequences. And adjusting continuity according to the preferences of models with Lipschitz Regularizer can largely improve their performances. More details of this experiment are in Appendix A.

two kinds of input sequences with different continuity, and map them to output sequences using exponential moving average. Then, we use two different deep models to learn this mapping. Each model has an identical 1D convolution embedding layer, and a separate sequence processing module. One uses the S4 model (Gu et al., 2021) and the other uses vanilla Transformer (Vaswani et al., 2017) (with the same number of layers and hidden dimensions). The results of this experiment are shown in Figure 1. It can be observed that the S4 model achieves significantly better performance with more continuous inputs, and the Transformer performs better with more discrete inputs. Note that, essentially, they are learning the same mapping only with different data continuity. This clearly shows different models prefer different data continuity.

Inspired by the above observation, we hypothesize that it is possible to enhance model performance by changing the data continuity according to their preferences. To make the proposed method simple and applicable for different deep models, we derive a surrogate that can be directly optimized for the Lipschitz continuity, and use it as a regularizer in the loss function. We call the proposed surrogate **Lipschitz Regularizer**, which depicts the data continuity and can also be used to adjust it.

Then, we investigate the data continuity property for different models and how to use Lipschitz Regularizer to change data continuity according to the model preference. We provide in-depth analyses in both time and frequency domains. On the one hand, Lipschitz continuity describes the continuity of sequences over time, which is a feature in the time domain. Here, we investigate two models. One is a continuous-time model S4, and the other model Informer is based on self-attention. As for S4, since the fitting error of the S4 model is bounded by the Lipschitz constant, S4 prefers smoother input sequences with smaller Lipschitz constant. Hence, we make the inputs of S4 layers more continuous by adding the Lipschitz Regularizer to the loss function. Experiment results on the Long Range Arena benchmark demonstrate that Lipschitz Regularizer can largely improve

the performance of the S4 model, especially for tasks with discrete inputs. Conversely, Informer is built upon self-attention, which is designed to process some tokenized discrete data, e.g., text, so Informer prefers less continuous sequences. Therefore, we decrease the continuity of input sequences by subtracting the Lipschitz Regularizer from the loss function. Prediction performance and empirical analyses on many time-series tasks well prove the superiority of the Lipschitz Regularizer. Also, we observe the same results on the task mentioned above as shown in Figure 1.

On the other hand, for the frequency domain, we find that Lipschitz Regularizer represents the expectation of the frequency of data's underlying function. Here, we take the ReLU network as the studied case, and theoretically justify that Lipschitz Regularizer is related to the spectral bias - a phenomenon that neural networks tend to prioritize learning the low-frequency modes. We then propose to use Lipschitz Regularizer by subtracting it from the loss function to mitigate the spectral bias. In this way, neural networks are forced to learn high-frequency parts, and convergence can be accelerated since information in different frequency bands can be learned simultaneously.

In summary, Lipschitz Regularizer can be used to flexibly adjust data continuity for a wide range of deep models which have a preference for data continuity. It improves various models with very little extra computational cost, shedding a light on inherent data property analyses for sequence modeling.

## 2 RELATED WORK

**Deep Neural Networks for Sequence Modeling**  Sequence modeling plays a critical role in many machine learning problems. Many general architectures, including MLPs (Rahaman et al., 2019), RNNs (Mikolov et al., 2010), and CNNs (Bai et al., 2018), can all be used for sequence modeling. And recently, two types of models show great power in addressing challenges for sequence modeling, such as handling complex interactions and long-range dependencies. The first type is self-attention-based models. For instance, the vanilla Transformer (Vaswani et al., 2017), Informer (Zhou et al., 2021), and Performer (Choromanski et al., 2020) all show great performance on natural language processing, time-series forecasting, and speech processing, respectively. Another type is continuous-time models, which are built upon the view that inputs are sampled from continuous functions. They include but not limited to Neural ODE (Chen et al., 2018), Lipschitz RNN (Erichson et al., 2020), State-space model (Gu et al., 2021). In this paper, we do not aim at proposing novel models like previous works, but we focus on understanding intrinsic preference for input sequences. We show these two types of models both have a preference for the data continuity property, and we utilize it to promote their performance.

**Lipschitz Continuity of Neural Networks**  The Lipschitz continuity is a general property for any function, and is widely used for analyzing different kinds of neural networks, including MLPs (Zhang et al., 2021; Gouk et al., 2021), CNNs (Zou et al., 2019), self-attention-based networks (Dasoulas et al., 2021), graph neural networks (Gama et al., 2020) and GANs (Gulrajani et al., 2017). It becomes an essential property of neural networks, and can be used in various ways, such as improving adversarial robustness (Meunier et al., 2022) and proving generalization bounds (Sokolić et al., 2017). In this paper, we focus on the Lipschitz continuity of the underlying function of sequence data, and use it as a data property, but not a property of models.

## 3 LIPSCHITZ CONTINUITY OF SEQUENCE DATA

In this section, we first show the measure for continuity of sequence data, and how it can be used as a regularizer. We give a definition for Lipschitz Regularizer here, and leave detailed analyzes and usages of it in specific models in the rest of the sections. Then, we provide views for Lipschitz Regularizer in both time and frequency domains.

To define the measure for the continuity of sequence data, we view inputs as signals, and data points in the sequence are discrete samples of an underlying continuous function with certain time steps. Next, we calculate the Lipschitz constant of the underlying function, which is widely used as the measure for continuity. Specifically, suppose the sequence is $x_0, x_1, \ldots, x_n$, and the underlying function is defined as $f(t_0) = x_0, f(t_1) = x_1, \ldots, f(t_n) = x_n$, where $t_0, t_1, \ldots, t_n$ are time steps.

Then, if we let $t_0 = 0, t_1 = 1, \ldots, t_n = n$, the Lipschitz constant $L_f$ of function $f$ is

$$L_f = \max_{t_i, t_j \in \{0,1,\ldots,n\}} \frac{|f(t_i) - f(t_j)|}{|t_i - t_j|} = \max_{i,j \in \{0,1,\ldots,n\}} \frac{|x_i - x_j|}{|i - j|}. \tag{1}$$

By Mean Value Theorem, for $i, j \in \{0, 1, \ldots, n\}$ and $j - i > 1$, we could always find an interval $[k, k+1]$ of time step 1 such that $i \le k \le j - 1$, $\frac{|x_i - x_j|}{|i - j|} \le |x_{k+1} - x_k|$. Therefore, we have

$$L_f = \max_{i,j \in \{0,1,\ldots,n\}} \frac{|x_i - x_j|}{|i - j|} = \max_{k \in \{0,1,\ldots,n-1\}} |x_{k+1} - x_k|. \tag{2}$$

However, since we would like to adjust this continuity according to the preferences of different models, this measure should be easy to be optimized, but it is hard to pass gradients due to the $\max$ operator. To help with the optimization process, we design a surrogate by taking the average over all terms and changing the norm to $\ell_2$. Moreover, since we simply use this surrogate as a regularizer in the loss function to flexibly adjust the continuity for various models, we name this term as *Lipschitz Regularizer*, and its formal definition is given as follows.

**Definition 3.1.** *(**Lipschitz Regularizer**) Suppose the sequence is $x_0, x_1, \ldots, x_n$. We define the Lipschitz Regularizer in the following equation:*

$$\mathcal{L}_{\text{Lip}} = \frac{1}{n} \sum_{i=0}^{n-1} (x_{i+1} - x_i)^2 \tag{3}$$

### 3.1 VIEW LIPSCHITZ REGULARIZER IN TIME AND FREQUENCY DOMAINS

We then provide two views for Lipschitz Regularizer. On the one hand, Lipschitz Regularizer is a feature for sequence data in the time domain, representing the continuity of sequences over time. Thus, it can be used to alter the continuity of input sequences to specific models. As shown in Figure 1, different deep models have different preferences for data continuity. We can use the Lipschitz Regularizer to manually make the sequences more or less continuous, and therefore improve the performance of the model. An example of increasing the continuity to improve the performance of a continuous-time model is described in §4.1. A converse example of decreasing the continuity to improve the performance of an attention-based model is described in §4.2.

On the other hand, from the frequency perspective, Lipschitz Regularizer directly relates to the frequency of the function, and can be used to alter modes with different frequencies. Specifically,

$$\sum_{i=0}^{n-1} (x_{i+1} - x_i)^2 \approx \int_{\mathbb{R}} \left( \frac{\mathrm{d}f(t)}{\mathrm{d}t} \right)^2 \mathrm{d}t = \int_{\mathbb{R}} (2\pi i \xi)^2 \, \hat{f}^2(\xi)(-\mathrm{d}\xi) = 4\pi^2 C \mathbb{E}_{p(\xi)}[\xi^2] \tag{4}$$

where $\xi$ is the frequency of the Fourier transform of $f$. $p(\xi) = \hat{f}^2(\xi)/C$ is the normalized squared Fourier transform of $f$, where $\hat{f}(\xi) := \int f(x) e^{-i2\pi\xi x} \mathrm{d}x$. Details of the derivation are presented in Appendix G.1. Essentially, the Lipschitz Regularizer of sequence data represents the exception of the frequency of the data's underlying function. Besides, previous literature shows that neural networks tend to prioritize the learning of low-frequency parts of the target function (Rahaman et al., 2019). We find that Lipschitz Regularizer can be utilized to mitigate this phenomenon by emphasizing high-frequency parts, which allows the network to fit all spectra simultaneously and results in a faster convergence rate. The details of this discussion are in §5.1.

## 4 TIME DOMAIN

In this section, we view Lipschitz Regularizer in the time domain, and show how it can be used to make the sequence more discrete or continuous over time, catering to the preference of different models. Generally, to change the continuity of the input sequence to different models with the Lipschitz Regularizer, we apply it on the output of the embedding layer before the sequence is sent to different models. We describe the details of two different models in the following sections.

Table 1: Accuracy of the S4 model and its variant with our proposed Lipschitz Regularizer (S4 + Emb + Lip) in LRA. S4 + Emb is set to ablate the effect of the extra embedding layer.

|  | ListOps | Text | Retrieval | Image | Image-c | Path | Path-c | PathX | PathX-c |
|---|---|---|---|---|---|---|---|---|---|
| S4 | 59.53 | 86.51 | 91.07 | 88.54 | 84.27 | **94.02** | 89.11 | **96.03** | 92.41 |
| S4 + Emb | 58.94 | 87.12 | 90.28 | 87.25 | 85.13 | 92.37 | 90.32 | 93.87 | 92.81 |
| **S4 + Emb + Lip** | **61.37** | **89.74** | **93.83** | **89.19** | **88.43** | 93.52 | **91.39** | 95.72 | **94.36** |

### 4.1 STATE SPACE MODEL

The State Space Model is a classic model in control engineering. Gu et al. (2021) extended it to the deep sequence model, and proposed the S4 model. S4 is a continuous-time sequence model. It advances SoTA on long-range sequence modeling tasks by a large margin. An S4 layer can be denoted as follows:

$$\dot{x} = Ax + Bu$$
$$y = Cx + Du, \tag{5}$$

where $u$ is the input function, $x$ is the hidden state , $y$ is the output. $A$, $B$, $C$, $D$ are trainable matrices.

The critical and essential design in the S4 layer is the transition matrix $A$, which is initialized with the HiPPO matrix. The HiPPO matrix makes the S4 layer optimally remember the history of the input's underlying function, so the S4 model can substantially outperform previous methods on long-range sequence modeling tasks. Particularly, the HiPPO matrix is designed to find the best polynomial approximation of the input's underlying function given a measure that defines the optimal history, and a memory budget which is the hidden dimension in the model. Each measure corresponds to an optimal HiPPO matrix.

**Theoretical Analyses** To connect the continuity property with the S4 model, we provide the following intuition here while a formal proposition along with its proof in Appendix G.2. Generally, the error rate of HiPPO-LegS projection decreases when the sequence is more continuous/smooth (Gu et al., 2021). Here, LegS denotes the scaled Legendre measure, which assigns uniform weights to all history. This is also true for S4 layers, since the HiPPO matrix is the most critical design in the S4 layer. However, in many tasks, such as natural language processing, the input sequence are not particularly smooth. This will deteriorate the performance of the S4 model.

Lipschitz Regularizer can be used to solve the above problem, because it can adjust the continuity of sequences. Specifically, since we cannot directly manipulate the underlying function of the input sequence, we add a 1D convolutional layer that does not change the sequence length as an embedding layer before the S4 layer, and then apply Lipschitz Regularizer to the output of the embedding layer as follows:

$$\mathcal{L}(y, \hat{y}, \hat{l}) = \mathcal{L}_{\text{S4}}(y, \hat{y}) + \lambda \mathcal{L}_{\text{Lip}}(\hat{l}), \tag{6}$$

where $y$ is the ground-truth, and $\hat{y}$ is the output of the S4 model. $\hat{l}$ is the output of the embedding layer, and $\mathcal{L}_{\text{S4}}$ is the original loss of the S4 model. $\lambda$ is a hyperparameter to control the magnitude of the Lipschitz Regularizer. By using Equation (6) as the loss function, the input of the S4 layers becomes more continuous, so the error of the HiPPO-LegS projection and S4 layer can be reduced, leading to better model performance.

**Experiments** To demonstrate the effectiveness of the Lipschitz Regularizer, we use a modified version of the Long Range Arena (LRA) (Tay et al., 2020) benchmark with harder tasks. The descriptions of the original LRA are in Appendix . In addition to the original LRA, we create 3 harder tasks with more discrete sequences. Particularly, we notice that among these 6 tasks, 3 of them use pixels as inputs (i.e., Image and Pathfinder), which could be more continuous than texts in the other 3 tasks. So we design *Image-c*, *Path-c*, and *PathX-c*, in which the contrast of images is increased, and the increasing degree is randomly sampled from 50% to 100% for each image.

We test three methods on the modified LRA. The first one is the original S4 model (denoted as *S4*). The second one is the S4 model with a 1D convolutional layer as the embedding layer, and Lipschitz Regularizer is applied to the outputs of the embedding layer (denoted as *S4 + Emb +*

*Lip*). Furthermore, we also design the third model to ablate the effect of the extra embedding layer. Here, we use the S4 model with the same embedding layer as the previous method, and Lipschitz Regularizer is not applied (denoted as *S4 + Emb*). Hyperparameter $\lambda$ is chosen from $\{1, 2, 3, 4, 5\}$ when the model performs best on the validation set.

Results are listed in Table 1, and we have the following observations. (1) It is obvious that our method (i.e., S4 + Emb + Lip) significantly outperforms other methods in almost all tasks, especially in tasks with discrete inputs, such as Text and Retrieval. Improved performance in Image-c, Path-c, and PathX-c shows that Lipschitz Regularizer can mitigate the influence of increased contrasts. These results well demonstrate the effectiveness of the Lipschitz Regularizer, indicating that it can make input sequences of the S4 layer more continuous, and better cater to the preference of the S4 model. (2) Comparing the results of S4 on Image/Path(X) and Image-c/Path(X)-c, it can be observed that the performance of the S4 model degenerates with the increasing contrasts of images. The cause is the deceased continuity against the preference of the S4 model, verifying that the S4 model indeed prefers continuous inputs. (3) Only adding the extra embedding layer (S4 + Emb) makes the accuracy decrease in 4 out of 7 tasks, indicating that improvements come from the effect of the Lipschitz Regularizer, but not the extra layer. Besides, this extra embedding layer is also the main reason causing the performance drop in Path and PathX dataset. In Appendix B.2, the visualization for the output vector of the embedding layer shows that this embedding layer may overly and incorrectly blur or even erase some informative shapes in the original picture, causing some necessary information lost, and the model confused.

## 4.2 TRANSFORMER-BASED MODELS

In this section, we show that Lipschitz Regularizer can improve the performance of Transformer-based models when inputs are continuous. In particular, we choose time-series forecasting tasks whose inputs are highly continuous, and we use three Transformer-based models, i.e. vanilla Transformer (Vaswani et al., 2017), Informer (Zhou et al., 2021) and Autoformer (Wu et al., 2021) to evaluate the effectiveness of Lipschitz Regularizer. Although these models already have a good performance on time-series forecasting tasks, due to the preference of Transformer-based models for discrete sequences (shown in Figure 1) and highly continuous inputs, we can still apply Lipschitz Regularizer to further improve the model by decreasing the continuity of input sequences. Specifically, since all three models have an embedding layer, we directly apply Lipschitz Regularizer to the output of the embedding layer as follows:

$$\mathcal{L}(y, \hat{y}, \hat{l}) = \mathcal{L}_{\text{Transformer}}(y, \hat{y}) - \lambda \mathcal{L}_{\text{Lip}}(\hat{l}), \tag{7}$$

where $y$ is the ground-truth, and $\hat{y}$ is the output of the respective model. $\hat{l}$ is the output of the embedding layer, and $\mathcal{L}_{\text{Transformer}}$ is the original loss of the Transformer-based model. $\lambda$ controls the magnitude of the Lipschitz Regularizer. Note that different from the usage in the S4 model, here we subtract Lipschitz Regularizer to make the input discrete, and cater to the model preference.

**Experiments** We use 5 datasets in this experiment and their descriptions are in Appendix C.1. Evaluation metrics are Mean Square Error (MSE) and Mean Absolute Error (MAE). Hyperparameter $\lambda$ is chosen from $\{1, 2, 3, 4, 5, 6, 7, 8\}$ when the model performs best on the validation set.

Results of *Transformer*, *Informer*, *Autoformer*, and these models with Lipschitz Regularizer (denoted as *Transformer + Lip*, *Informer + Lip* and *Autoformer + Lip*) are shown in Table 2, and results of multivariate experiments are in Appendix C.2. We can see that the models with Lipschitz Regularizer generally outperform the original models on most of the tasks. This indicates that Transformer-based models prefer discrete sequences and reducing input continuity with Lipschitz Regularizer can be helpful for them. We also note that the Lipschitz Regularizer is more effective on vanilla Transformer than the models with specialized designs for time series forecasting. This indicates the vanilla Transformer is more sensitive to data continuity, and special designs in Informer and Autoformer may mitigate it.

Besides, to investigate whether Lipschitz Regularizer changes the data continuity, we also show curves tracking the Lipschitz constant of the output of the embedding layer during training in Figure 2. Curves show that with the Regularizer, the Lipschitz constant becomes larger than it is in the original model, and continuously increases during training. Results demonstrate that Lipschitz Regularizer can indeed change the continuity, and thus improve the model.

Table 2: Univariate time-series forecasting results of 3 Transformer-based models and training them with Lipschitz Regularizer (indicated by + Lip). Note in this table, prediction window sizes are converted to lengths of sequences used in the model.

| Methods | Transformer | | Transformer + Lip | | Informer | | Informer + Lip | | Autoformer | | Autoformer + Lip | |
|---|---|---|---|---|---|---|---|---|---|---|---|---|
| Metric | MSE | MAE | MSE | MAE | MSE | MAE | MSE | MAE | MSE | MAE | MSE | MAE |
| ETTh₁ 24 | 0.07047 | 0.20586 | **0.07019** | **0.20570** | 0.09842 | 0.24747 | **0.08882** | **0.23674** | 0.05567 | 0.18596 | **0.05504** | **0.18495** |
| ETTh₁ 48 | 0.18902 | 0.37046 | **0.16716** | **0.34974** | 0.15845 | 0.31907 | **0.12615** | **0.28333** | 0.07860 | 0.22324 | **0.07422** | **0.21398** |
| ETTh₁ 168 | 0.39773 | 0.55569 | **0.30811** | **0.48183** | 0.18314 | 0.34619 | **0.10579** | **0.25552** | 0.09232 | 0.24037 | **0.08983** | **0.23544** |
| ETTh₁ 336 | 0.41523 | 0.56902 | **0.41324** | **0.56402** | 0.22164 | 0.38720 | **0.11810** | **0.26959** | 0.10462 | 0.25484 | **0.10461** | **0.25483** |
| ETTh₁ 720 | 0.65586 | 0.75324 | **0.62233** | **0.73160** | 0.26883 | 0.43506 | **0.13131** | **0.28731** | **0.12069** | **0.27791** | 0.12394 | 0.27833 |
| ETTh₂ 24 | 0.09449 | 0.24259 | **0.07560** | **0.20989** | 0.09309 | 0.24015 | **0.08626** | **0.22559** | 0.11136 | 0.26315 | **0.09345** | **0.25515** |
| ETTh₂ 48 | 0.15016 | 0.30996 | **0.13229** | **0.29278** | 0.15483 | 0.31445 | **0.13684** | **0.28936** | 0.15137 | 0.30316 | **0.14945** | **0.30129** |
| ETTh₂ 168 | 0.25197 | 0.41087 | **0.21046** | **0.37453** | **0.23193** | **0.38947** | 0.30071 | 0.43671 | 0.20403 | 0.35646 | **0.18370** | **0.33714** |
| ETTh₂ 336 | 0.22258 | 0.38170 | **0.20867** | **0.37298** | 0.26321 | 0.41659 | **0.24875** | **0.40827** | 0.22188 | 0.37417 | **0.21195** | **0.36425** |
| ETTh₂ 720 | 0.21932 | 0.38844 | **0.18445** | **0.35793** | 0.27722 | 0.43063 | **0.23646** | **0.39648** | 0.25612 | 0.40089 | **0.25604** | **0.40085** |
| ETTm₁ 24 | 0.01279 | 0.08410 | **0.01210** | **0.08312** | 0.03016 | 0.13717 | **0.01815** | **0.09147** | 0.02317 | 0.11778 | **0.02300** | **0.10107** |
| ETTm₁ 48 | 0.08974 | 0.25869 | **0.02872** | **0.12820** | 0.06944 | 0.20255 | **0.05848** | **0.19686** | 0.04130 | 0.15783 | **0.03931** | **0.15601** |
| ETTm₁ 96 | 0.05341 | 0.17696 | **0.05182** | **0.15017** | 0.19414 | 0.37236 | **0.13336** | **0.30091** | 0.05432 | 0.18033 | **0.05258** | **0.17605** |
| ETTm₁ 288 | 0.22354 | 0.40455 | **0.13780** | **0.29825** | 0.40140 | 0.55355 | **0.30266** | **0.46864** | 0.11893 | 0.27181 | **0.07521** | **0.21728** |
| ETTm₁ 672 | **0.40726** | 0.55824 | **0.40726** | 0.55826 | 0.51164 | 0.64390 | **0.27543** | **0.45377** | **0.09156** | **0.23690** | 0.09280 | 0.23621 |
| Weather 24 | 0.00223 | 0.03468 | **0.00154** | **0.02497** | 0.11676 | 0.25142 | **0.11256** | **0.23844** | 0.00740 | 0.06422 | **0.00736** | **0.06329** |
| Weather 48 | 0.00422 | 0.04106 | **0.00292** | **0.03026** | **0.17822** | **0.31846** | 0.19134 | 0.32408 | 0.01002 | **0.07648** | **0.00978** | 0.07727 |
| Weather 168 | 0.00537 | 0.05975 | **0.00319** | **0.04464** | 0.26585 | 0.39764 | **0.25138** | **0.37400** | 0.01038 | 0.07082 | **0.00528** | **0.05638** |
| Weather 336 | 0.00524 | 0.05772 | **0.00417** | **0.03673** | 0.29713 | 0.41571 | **0.24748** | **0.37725** | 0.00729 | 0.06492 | **0.00566** | **0.05888** |
| Weather 720 | 0.00933 | 0.07630 | **0.00272** | **0.03823** | 0.35875 | 0.46647 | **0.26479** | **0.39214** | 0.00960 | 0.08758 | **0.00925** | **0.07136** |
| ECL 48 | 0.26161 | 0.37762 | **0.24277** | **0.36460** | 0.23894 | 0.35891 | **0.17306** | **0.30488** | 0.57845 | 0.55653 | **0.52893** | **0.54479** |
| ECL 168 | 0.32283 | 0.41766 | **0.29221** | **0.39677** | 0.44680 | 0.50315 | **0.18765** | **0.31497** | 0.52339 | 0.53251 | **0.42961** | **0.48926** |
| ECL 336 | 0.47213 | 0.50381 | **0.37916** | **0.45903** | 0.48892 | 0.52840 | **0.37758** | **0.43689** | 0.53511 | 0.54372 | **0.51762** | **0.52391** |
| ECL 720 | 0.48477 | 0.52413 | **0.46524** | **0.51649** | 0.54026 | 0.57059 | **0.44496** | **0.50387** | 0.80028 | 0.66756 | 0.91144 | 0.69692 |
| ECL 960 | 0.46930 | 0.51409 | **0.43643** | **0.49698** | 0.58225 | 0.60782 | **0.37740** | **0.45817** | 0.76603 | 0.65389 | **0.64859** | **0.62054** |
| Count | 2 | | 49 | | 4 | | 46 | | 6 | | 44 | |

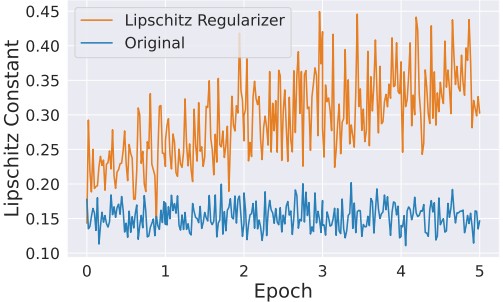

Figure 2: The Lipschitz constant of the output of the embedding layer during the training process of Informer + Lip. The experiment is the univariate ETTh₂ with the prediction window size of 24h.

Figure 3: MSE of Informer + Lip on univariate EETh₂ with different $\lambda$. Colors represent different window sizes.

We also show an ablation study for the hyperparameter $\lambda$ in Figure 3. We can observe that (1) MSE increases when $\lambda < 0$, while decreases when $\lambda > 0$. Since positive $\lambda$ reduces the data continuity, we can conclude that Informer prefers discrete sequences, and Lipschitz Regularizer can reduce the continuity and cater to the preference; (2) MSEs do not have a large variance for different positive $\lambda$, indicating that the performance improvement is not sensitive to hyperparameter changes.

## 5 FREQUENCY DOMAIN

In this section, we study how continuity affects the performance of deep models from the frequency perspective. We take the ReLU network as the study case, and provide theoretical analyses and experiment results to show the effectiveness of applying the Lipschitz Regularizer on ReLU networks.

## 5.1 RELU NETWORK

A ReLU network $g : \mathbb{R}^d \to \mathbb{R}$ with $L$ hidden layers of width $d_1, \ldots, d_L$ is defined as:

$$g(\mathbf{x}) = \left( T^{(L+1)} \circ \sigma \circ T^{(L)} \circ \cdots \circ \sigma \circ T^{(1)} \right)(\mathbf{x}), \tag{8}$$

where $T^{(k)} : \mathbb{R}^{d_{k-1}} \to \mathbb{R}^{d_k}$ is an affine function ($d_0 = d$, $d_{L+1} = 1$) and $\sigma$ is the ReLU function.

**Theoretical Analyses** In the previous literature, Rahaman et al. (2019) showed that the low-frequency part of the sequence data is learned faster by the ReLU network, and such phenomenon is called the "spectral bias". We claim that the Lipschitz Regularizer could help mitigate the spectral bias. Intuitively, when the Lipschitz constant of the ReLU network increases, we expect that the model can learn more information in high-frequency parts. We provide a formal proposition and its proof on this intuition in Appendix G.3. This inspires us to balance frequency modes via changing the Lipschitz continuity of functions. Besides, suppose we use a ReLU network to learn a sequence-to-sequence mapping, where values of data in the input sequence (length $n$) increase linearly in the interval $(0, 1)$ with step size $\frac{1}{n}$, and the output is generated by the mapping function $h(t)$. Note that since values of data in the input sequence increase linearly, the Lipschitz constant of the ReLU network is the same as the output sequence. Therefore, we design the decayed Lipschitz Regularizer as follows:

$$\mathcal{L}(y, \hat{y}) = \mathcal{L}_{\mathrm{MSE}}(y, \hat{y}) - \lambda e^{-\epsilon t} \mathcal{L}_{\mathrm{Lip}}(\hat{y}), \tag{9}$$

where $y$ is the ground-truth generated by $h(t)$ and $\hat{y}$ is the prediction. $\mathcal{L}_{\mathrm{MSE}}$ is the MSE Loss. $\lambda$ and $\epsilon$ are hyperparameters that control the magnitude and decay rate of the Lipschitz Regularizer, respectively.

We further explain the reasons why this regularizer can mitigate spectral bias in two aspects. First, by Equation (4), the added term could be seen as a direct penalty to the low-frequency part of the output sequence. Since the value of data in the input sequence increases linearly, this is equivalent to penalizing the low-frequency part of the ReLU network, and prioritizing the learning of the high-frequency part.

In another perspective, Rahaman et al. (2019) claimed that the origin of the spectral bias is the gradually increasing parameter norm, and Lipschitz Regularizer can intentionally relieve it. Specifically, Fourier components of the ReLU network $\hat{g}_\theta(\xi)$ is bounded by $O(L_g)$, and $L_g$ is bounded by the parameter norm, which can only increase by a small step during the optimization step. Hence, gradually increasing parameter norms can hinder the learning of high-frequency parts at the early optimization stage. Besides, due to the fact that Lipschitz Regularizer can intentionally change $L_g$, subtracting Lipschitz Regularizer as Equation (9) can enlarge the parameter norm, making it possible for optimizing both high and low-frequency parts. This can be seen as a warm-up process for the network where the parameter norm increases at the beginning of optimization, and then the convergence can be significantly accelerated, since modes of all frequencies can be learned simultaneously after the warm-up.

**Experiments** We choose a mapping task to evaluate the proposed Lipschitz Regularizer. Specifically, we try to learn the mapping function whose input is the sequence with linearly increasing values, and output is a highly periodic sequence. Given frequencies $K = \{k_1, k_2, \ldots, k_n\}$, amplitudes $A = \{a_1, a_2, \ldots, a_n\}$, and phases $\Phi = \{\phi_1, \phi_2, \ldots, \phi_n\}$, the mapping function is defined as $h(x) = \sum_{i=1}^{n} A_i \sin(2\pi k_i x + \phi_i)$.

In this experiment, we take $n = 10$ and frequencies $K = \{5, 10, \ldots, 45, 50\}$, amplitudes $A = \{0.1, 0.2, \ldots, 1\}$. The phases are uniformly sampled from 0 to $2\pi$, i.e., $\phi_i \sim U(0, 2\pi)$. The input samples in the sequence are uniformly placed over $(0, 1)$ with the number of samples $N = 100$, and the output is generated by $h(x)$.

As for the model, we use a 6-layer deep ReLU network with the hidden dimension set to 256 for all layers. To verify the effectiveness of the proposed Lipschitz Regularizer, we train two identical networks with the same training procedure. One is trained with the decayed Lipschitz Regularizer and the other without it. We set hyperparameter $\lambda \in \{1, 2, 3, 4, 5\}$ and $\epsilon \in \{0.00001, 0.0001, 0.001, 0.01, 0.1\}$ when the model performs best on the validation set.

We show the frequency and MSE of ReLU networks during the training process in Figure 4. From Figure 4 (a) and (b), we notice that low-frequency parts are learned first in both networks, but

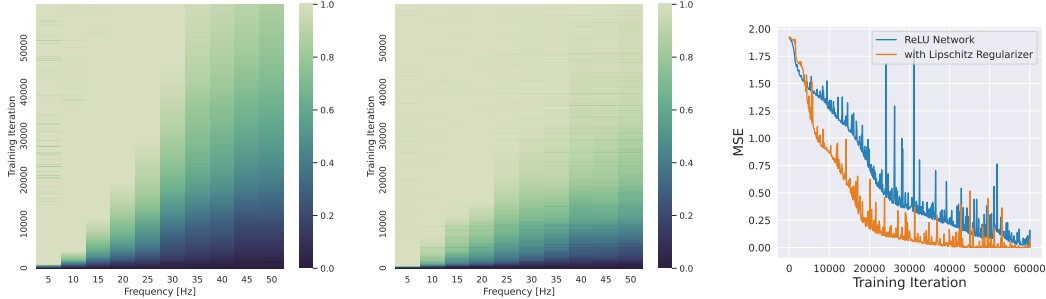

(a) Frequency of a ReLU network

(b) Frequency of an identical ReLU network with Lipschitz Regularizer

(c) MSE of the ReLU network w and w/o Lipschitz Regularizer

Figure 4: Evolution of the frequency and MSE of ReLU networks during the training process. In (a) and (b), color indicates the normalized amplitude of the Fourier component at the corresponding frequency, i.e., $|\hat{g}_\theta(k_i)|/A_i$. Lipschitz Regularizer enables faster learning of high frequencies and faster convergence.

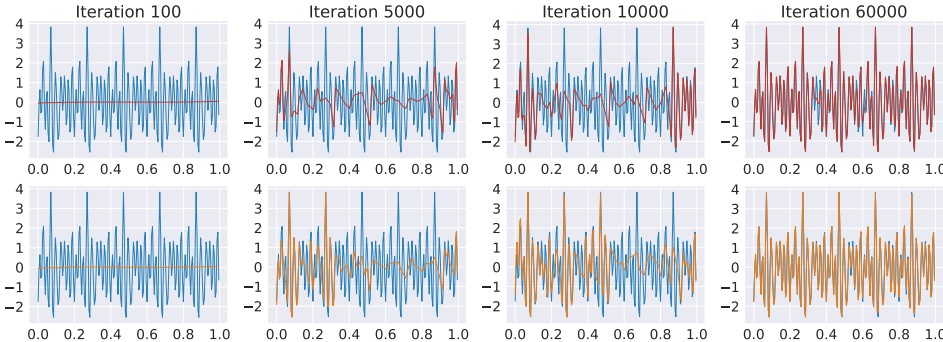

Figure 5: Predictions of ReLU networks (first row: without Lipschitz Regularizer; second row: with Lipschitz Regularizer) during the training process. With Lipschitz Regularizer, high-frequency parts can be learned faster.

with the decayed Lipschitz Regularizer, high frequencies can be learned significantly faster. Figure 4 (c) shows that Lipschitz Regularizer can accelerate convergence. We also show predictions of two models during the training process in Figure 5, which gives a more intuitive result, indicating that high frequencies can be learned faster when we use decayed Lipschitz Regularizer to warm-up optimization. All results demonstrate that Lipschitz Regularizer enables almost simultaneous learning for all frequencies, so spectral bias can be relieved in this way, and the convergence is accelerated.

## 6 SUMMARY

We investigate a generic property of sequence data, i.e., *continuity*, which is closely related to the performance of different models, and propose *Lipschitz Regularizer* to flexibly adjust the continuity for various models. We first empirically observe that the different deep models prefer different data continuity. Then, from both time and frequency domains, we provide in-depth theoretical and experimental studies for specific models. For the time domain, we show that the continuous-time model S4 prefers continuous sequences, while the Transformer-based model Informer prefers discrete inputs. We use Lipschitz Regularizer to adjust the data continuity for both of them and largely improve their performance by catering to their preference. For the frequency domain, we show that Lipschitz Regularizer can help mitigate the spectral bias, and accelerate convergence for ReLU networks. In general, Lipschitz Regularizer is available for any sequence modeling tasks and models which have a preference for data continuity, and can accordingly facilitate learning for various models with very little computational cost.

ACKNOWLEDGMENTS

The authors would like to thank Yifei Shen and Yansen Wang for their helpful discussions and insights. The authors also want to thank our reviewers for providing all the valuable feedback and suggestions.

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

# APPENDIX

## TABLE OF CONTENTS

## A  DETAILS OF EXPERIMENT IN THE INTRODUCTION

### A.1  UNIVARIATE

We present a sequence-to-sequence task in the Introduction section, and show more details here. In this experiment, we generate two types of input sequences with different continuity (each has 1000 samples), and map them to output with the exponential moving average $h(t)$:

$$h(t) = \frac{2}{1+w} x_t + \left(1 - \frac{2}{1+w}\right) h(t-1)$$

where $x_1, x_2, \ldots, x_N$ is the input sequence, $w$ is the window size (set to 50). We choose the exponential moving average because it is a sequence-to-sequence mapping that makes use of the contextual information. The Lipschitz constant of the input and output sequence is shown in Table 3. Note that the high Lipschitz constant represents low continuity, while the low Lipschitz constant represents high continuity. Then, we train the S4 model and the Transformer model with generated input and output sequences. Each model has a 1D convention embedding layer with kernel size 5, stride 1, and padding 2. Both Transformer and S4 have 1 separated layer with the hidden dimension set to 16. We also apply Lipschitz Regularizer to the output of the embedding layer and train models again. MSE of these 4 models is shown in Table 3. We could observe that S4 performs better with continuous inputs and the Transformer is better with discrete inputs. Also, Lipschitz Regularizer can improve the performance of S4 and Transformer by changing the data continuity into their prefers ones.

### A.2  MULTIVARIATE

We repeat the above experiment with multivariate data. Specifically, we also generate high and low continuity input sequences with dimension 16 (each has 1000 samples). The input sequences are

Table 3: Results of the experiment in the Introduction.

|  |  | High Continuity | Low Continuity |
|---|---|---|---|
| Lipschitz Constant | Input | 0.0543 | 0.2706 |
|  | Output | 0.0107 | 0.0142 |
| MSE | S4 | 0.00014 | 0.00567 |
|  | S4 + Lip | - | 0.00045 |
|  | Transformer | 0.00003 | 0.00222 |
|  | Transformer + Lip | 0.00038 | - |

mapped to output with exponential moving average. Same models are trained on the generated data. Results are shown in Table 4. We also randomly sample four dimensions and corresponding curves are shown in Figure 6, 7, 8, 9. Our findings and conclusions in the univariate experiment also hold in the multivariate case.

Table 4: Results of the experiment in the Introduction running with multivariate data.

|  |  | High Continuity | Low Continuity |
|---|---|---|---|
| Lipschitz Constant | Input | 0.0708 | 0.3592 |
|  | Output | 0.0138 | 0.0171 |
| Average MSE over all variates | S4 | 0.00072 | 0.00963 |
|  | S4 + Lip | - | 0.00141 |
|  | Transformer | 0.00014 | 0.01368 |
|  | Transformer + Lip | 0.00418 | - |

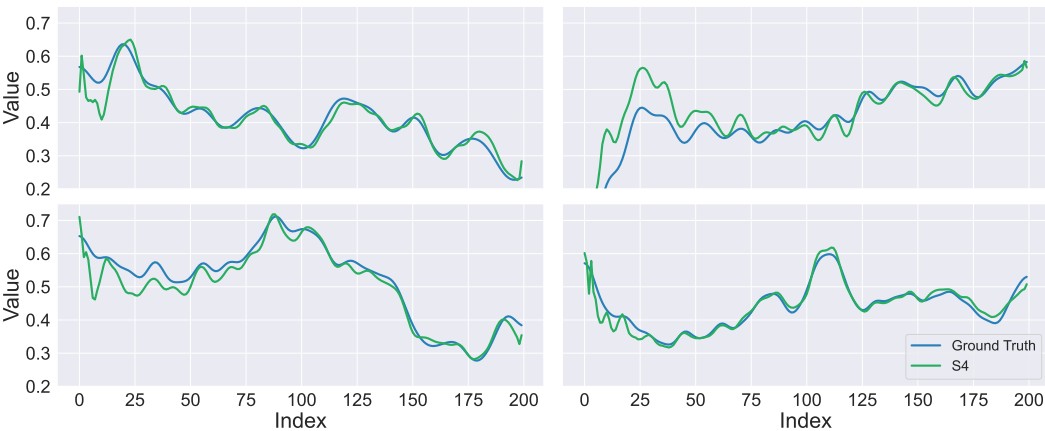

Figure 6: Results of the S4 model with high continuity multivariate data for the experiment in the Introduction.

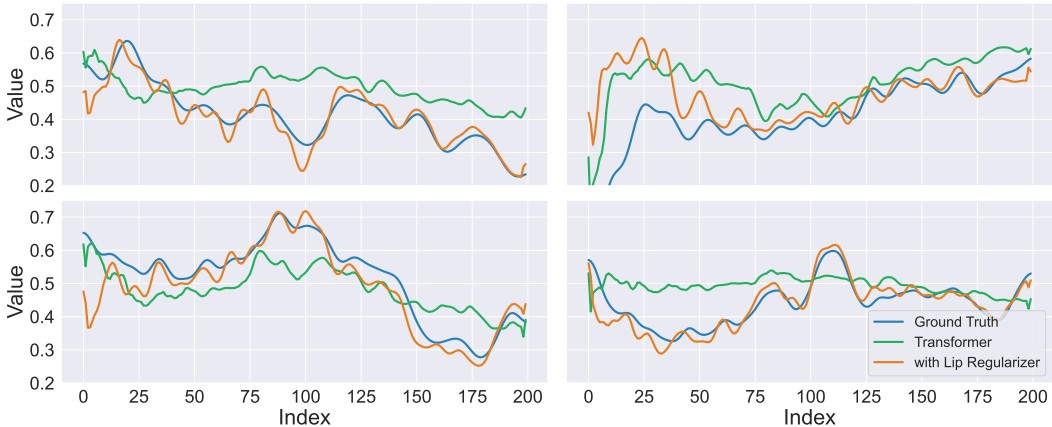

Figure 7: Results of the Transformer model with high continuity multivariate data for the experiment in the Introduction.

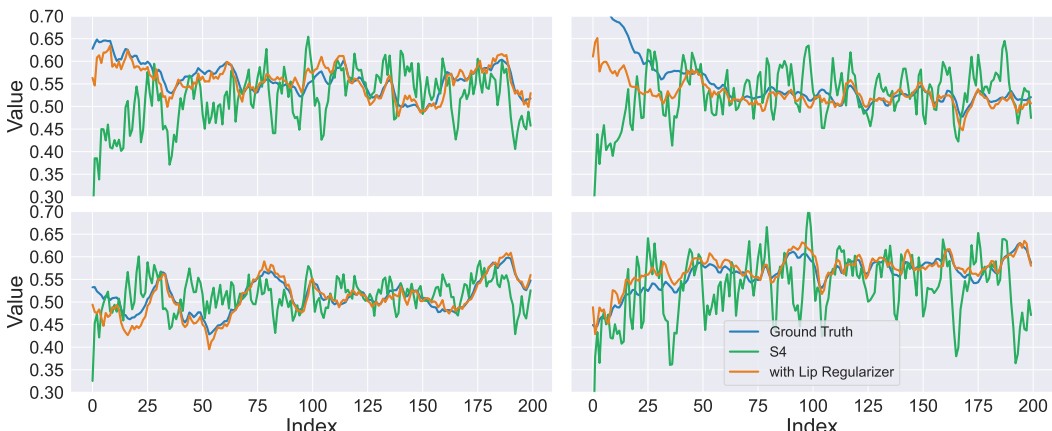

Figure 8: Results of the S4 model with low continuity multivariate data for the experiment in the Introduction.

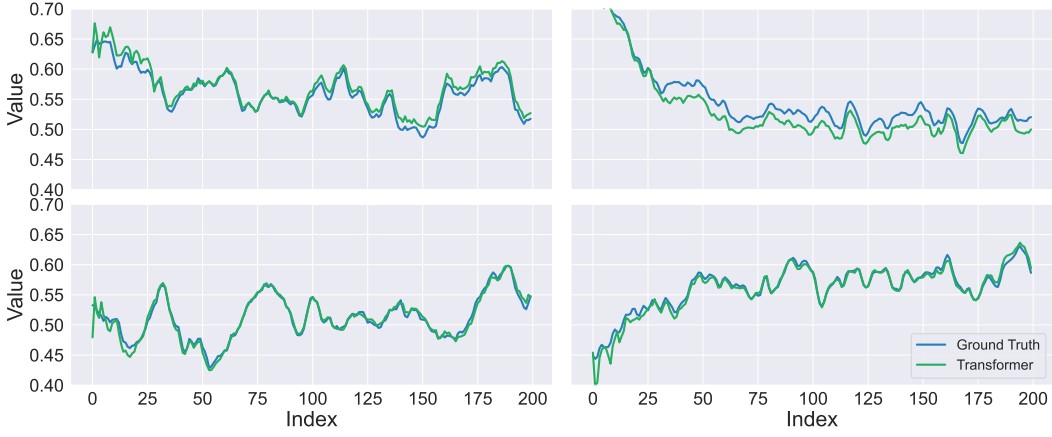

Figure 9: Results of the Transformer model with low continuity multivariate data for the experiment in the Introduction.

# B    DETAILS OF EXPERIMENT FOR THE S4 MODEL

## B.1    DATASET DESCRIPTION

In this experiment, we test the Lipschitz Regularizer on the S4 model. The testing dataset is the Long Range Arena (LRA) benchmark. Specifically, LRA contains various long sequence modeling tasks with sequence lengths ranging from 1K to 16K, which is very challenging for deep models. Tasks in LRA include (1) *ListOps*, in which the model needs to calculate the answer with a list of algebraic operations, (2) *Text*, a binary classification task with byte-level texts from IMDB reviews, (3) *Retrieval*, a document retrieval task with byte-level texts and documents from ACL Anthology Network, (4) *Image*, an image classification task with the image in CIFAR-10 dataset flattened into the pixel sequence, (5) *Path* (referred as Pathfinder in Gu et al. (2021)), in which the model needs to deduce whether two points in the image are connected by dashed lines, and the image is also flattened into the pixel sequence, (6) *PathX*, a harder version of *Path*, where the dimension of input images increased from $32 \times 32$ to $128 \times 128$.

## B.2    ANALYSIS FOR THE PATHFINDER TASK

We notice that the Lipschitz Regularizer causes deteriorated performance on the Pathfinder dataset, so here, we provide detailed analyzes to explore the reason. We show a case in the experiment of the S4 model on the Path dataset in Figure 10. We can see that the performance drop is mainly caused by the embedding layer. As explained in § 4.1, since we cannot directly manipulate the underlying function of the input sequence, we add an extra embedding layer before the S4 layer. However, changes from figure a1 to a2 show that this embedding layer may overly and incorrectly blur or even erase some informative shapes in the original picture, causing some necessary information lost, and making the model confused. Although Lipschitz Regularizer can slightly relieve this issue, the necessary path information is not as obvious as it is in the original image. The performance of these 3 models (i.e., S4, S4 + Emb, S4 + Emb + Lip) in Table 1 also matches this finding. Moreover, Figure b1, b2, and b3 show that when the contrast is increased, these shapes are not likely to be erased since their pixels all have high gray values. Hence, Lipschitz Regularizer can improve model performance on the Path-c task.

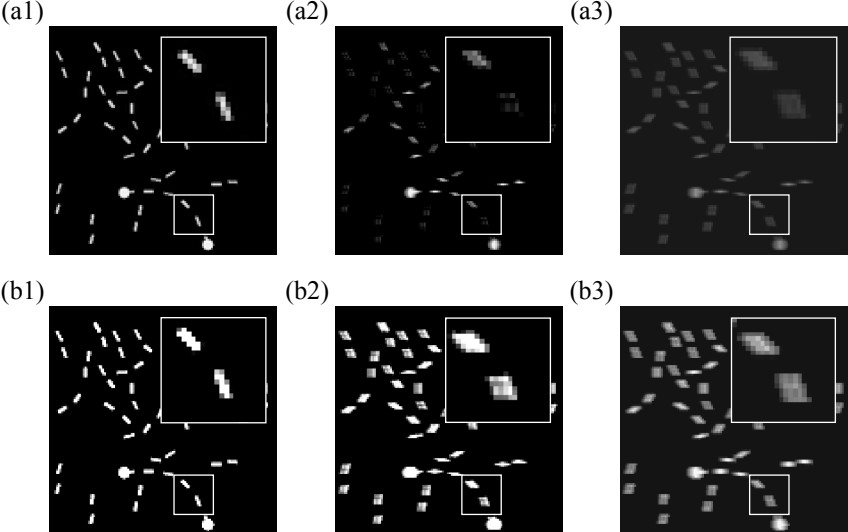

Figure 10: A case in the experiment of the S4 model on the Path dataset. In this task, the model needs to deduce whether two points in the image are connected by a dashed line. (a1) An image randomly sampled from the Path dataset. (b1) The image in (a1) with 100% contrast increased. (a2, b2) Average of the output vector of the embedding layer in a trained S4 + Emb model, with a1 and b1 as the input, respectively. (a3, b3) Average of the output vector of the embedding layer in a trained S4 + Emb + Lip model, with a1 and b1 as the input, respectively.

## C  DETAILS OF EXPERIMENT FOR THE TRANSFORMER-BASED MODEL

### C.1  DATASET DESCRIPTION

In this experiment, we show that the Lipschitz Regularizer can improve the performance of Transformer-based models, including Transformer, Informer, and Autoformer. We use 3 real-world datasets: ETT (Electricity Transformer Temperature), ECL (Electricity Consuming Load), and Weather. ETT has 3 separate datasets, i.e., $\{ETTh_1, ETTh_2\}$ for 1-hour-level with 2 separated countries, and $ETTm_1$ for 15-minute-level. And we use multiple prediction window sizes, including $\{24h, 48h, 168h, 336h, 720h, 960h\}$ for ETTh, ECL, Weather, and $\{6h, 12h, 24h, 72h, 168h\}$ for ETTm.

### C.2  RESULTS OF MULTIVARIATE TIME-SERIES FORECASTING

Results of multivariate time-series forecasting of three Transformer-based models are presented in Table 5. We can see that Lipschitz Regularizer can improve the model performance in most cases, showing that altering data continuity is also helpful in multivariate time-series forecasting tasks. Besides, we can also observe that the improvement by Lipschitz Regularizer is slightly less significant than which in univariate time-series forecasting. The reason may be that we apply the same regularizer to input sequences of all variates. However, these input sequences may have different continuities and need regularizers with different weights. Future work might be adding trainable weights to the regularizer of different input sequences.

Table 5: Multivariate time-series forecasting results of 3 Transformer-based models and training them with Lipschitz Regularizer (indicated by + Lip). Note in this table, prediction window sizes are converted to lengths of sequences used in the model.

| Methods | | Transformer | | Transformer + Lip | | Informer | | Informer + Lip | | Autoformer | | Autoformer + Lip | |
|---|---|---|---|---|---|---|---|---|---|---|---|---|---|
| Metric | | MSE | MAE | MSE | MAE | MSE | MAE | MSE | MAE | MSE | MAE | MSE | MAE |
| ETTh$_1$ | 24 | 0.59446 | 0.57293 | **0.57268** | **0.56055** | 0.57727 | 0.54945 | **0.53483** | **0.51756** | 0.39562 | 0.43466 | **0.37109** | **0.40143** |
| | 48 | 0.82961 | 0.70984 | **0.71223** | **0.65060** | 0.68461 | 0.62487 | **0.68292** | **0.61618** | 0.41434 | 0.44104 | **0.38277** | **0.41764** |
| | 168 | 1.05050 | **0.83959** | 1.04581 | 0.84110 | **0.93119** | **0.75159** | 1.02153 | 0.79883 | 0.46037 | **0.46466** | 0.46020 | 0.47120 |
| | 336 | 1.40297 | 0.99042 | **1.04308** | **0.80437** | 1.12811 | 0.87302 | **0.99066** | **0.75376** | 0.51710 | 0.49936 | **0.50278** | **0.48853** |
| | 720 | 1.05294 | 0.80128 | **1.04062** | **0.80127** | 1.21454 | 0.89606 | **1.20265** | **0.88891** | 0.47690 | 0.49172 | 0.50977 | 0.50879 |
| ETTh$_2$ | 24 | 0.83025 | 0.71355 | **0.31699** | **0.42514** | 0.72025 | 0.66539 | **0.33658** | **0.43113** | 0.26384 | 0.34252 | **0.25465** | **0.33327** |
| | 48 | 1.28728 | 0.92881 | **0.65413** | **0.64771** | 1.45708 | 1.00062 | 1.81144 | 1.08344 | 0.31247 | 0.37143 | **0.31090** | **0.37014** |
| | 168 | 5.66520 | 1.91462 | **3.56351** | **1.45387** | 3.48945 | 1.51457 | **3.05333** | **1.44302** | **0.46657** | 0.46249 | 0.46982 | **0.46203** |
| | 336 | 5.11314 | 1.81923 | **2.95696** | **1.37183** | 2.72290 | 1.33987 | **2.35546** | **1.28684** | **0.47924** | **0.48042** | 0.49315 | 0.48750 |
| | 720 | 3.00465 | 1.44033 | **2.52934** | **1.26787** | **3.46729** | **1.47321** | 3.72312 | 1.66671 | 0.47352 | 0.48400 | **0.46775** | **0.48066** |
| ETTm$_1$ | 24 | **0.28475** | 0.35112 | 0.28900 | **0.34211** | 0.32315 | 0.36881 | 0.35075 | 0.39111 | **0.35616** | 0.40320 | 0.35807 | **0.40252** |
| | 48 | 0.45711 | 0.45664 | **0.43928** | **0.44483** | 0.49426 | 0.50311 | **0.44407** | **0.45275** | 0.44404 | 0.44776 | **0.43239** | **0.44631** |
| | 96 | 0.68831 | 0.60444 | **0.53434** | **0.51490** | 0.67758 | 0.61353 | **0.47372** | **0.47680** | 0.55732 | 0.49872 | **0.55357** | **0.49706** |
| | 288 | 0.88883 | 0.71069 | **0.83600** | **0.66682** | 1.05643 | 0.78565 | **1.03285** | **0.80819** | 0.58302 | 0.50759 | **0.56224** | **0.50469** |
| | 672 | 1.17478 | 0.83787 | **0.92590** | **0.73367** | 1.19203 | 0.92626 | **0.90468** | **0.72072** | 0.56261 | 0.50822 | **0.56208** | **0.50788** |
| Weather | 24 | **0.14897** | **0.23293** | 0.21319 | 0.28302 | 0.33501 | **0.38091** | 0.32660 | 0.38280 | **0.15944** | **0.24224** | 0.16573 | 0.24907 |
| | 48 | 0.26360 | 0.33987 | **0.22625** | **0.31913** | 0.39546 | 0.45890 | **0.38138** | **0.43057** | 0.21638 | 0.29847 | **0.21580** | **0.29788** |
| | 168 | 0.49273 | 0.48432 | **0.42120** | **0.41574** | 0.60843 | 0.56714 | **0.60603** | **0.56406** | **0.30901** | **0.37048** | 0.31191 | 0.37294 |
| | 336 | 0.66636 | 0.58604 | **0.60194** | **0.57205** | 0.70204 | **0.61955** | **0.67344** | 0.62094 | 0.35603 | 0.39652 | **0.34620** | **0.38616** |
| | 720 | 0.89504 | 0.69045 | **0.50826** | **0.52560** | 0.83106 | 0.73079 | **0.65801** | **0.60372** | 0.42178 | 0.43347 | **0.42083** | **0.43125** |
| ECL | 48 | 0.24832 | 0.35046 | **0.23187** | **0.33168** | 0.34399 | 0.39289 | **0.25829** | **0.35700** | 0.18320 | 0.29892 | **0.18281** | **0.29758** |
| | 168 | 0.26446 | 0.36343 | **0.23862** | **0.33770** | 0.36820 | 0.42427 | **0.37272** | 0.43410 | 0.22662 | 0.33740 | **0.21555** | **0.32441** |
| | 336 | 0.27420 | 0.37078 | **0.25691** | **0.36375** | 0.38061 | 0.43101 | **0.34641** | **0.42147** | 0.22740 | 0.33667 | 0.23392 | 0.34069 |
| | 720 | 0.28641 | 0.37610 | **0.27986** | **0.36451** | 0.40616 | 0.44335 | **0.35887** | **0.43815** | 0.29826 | 0.38835 | **0.36776** | **0.33395** |
| | 960 | 0.32819 | 0.39887 | **0.30984** | **0.38930** | 0.45993 | 0.54774 | **0.44522** | **0.50400** | 0.27856 | 0.37727 | **0.23146** | **0.31658** |
| Count | | 4 | | 46 | | 12 | | 38 | | 13 | | 38 | |

### C.3  CASES IN TIME-SERIES FORECASTING TASKS

We present some cases to provide some deeper analyses for Lipschitz Regularizer. Here, we plot the last dimension of forecasting results for a qualitative comparison. In all figures of this section, the first 96 data points are inputs of the model, and others are forecasting results.

**Univariate Forecasting**  We show three examples of univariate forecasting in Figure 11, 12, 13. Figure 11 and 12 show how the Lipschitz Regularizer improves the performance of the model, while

Figure 13 shows a negative case. In Figure 13, the main problem is that the model trained with Lipschitz Regularizer captures a larger decrease than the original data when the time is in 100-200.

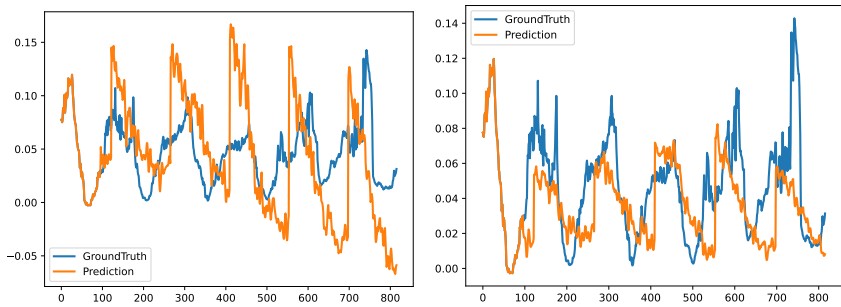

Figure 11: Univariate forecasting example of Transformer on the Weather dataset with the prediction window size set to 720. Left figure shows the result of the original Transformer (MSE: 0.00933, MAE: 0.07630). Right figure shows the result of the Transformer trained with Lipschitz Regularizer ($\lambda = 1$, MSE: 0.00272, MAE: 0.03823).

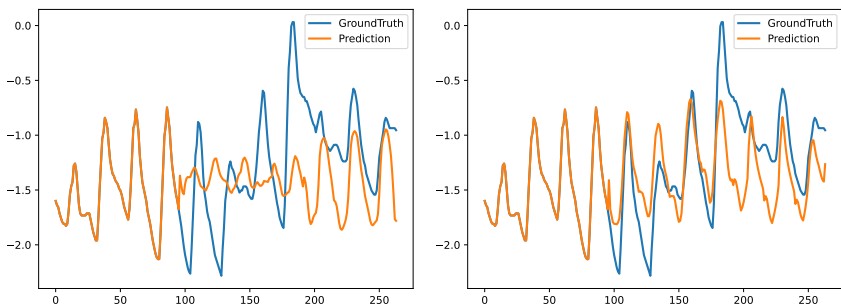

Figure 12: Univariate forecasting example of Autoformer on the ETTh2 dataset with the prediction window size set to 168. Left figure shows the result of the original Autoformer (MSE: 0.20403, MAE: 0.35646). Right figure shows the result of the Autoformer trained with Lipschitz Regularizer ($\lambda = 5$, MSE: 0.18370, MAE: 0.33714).

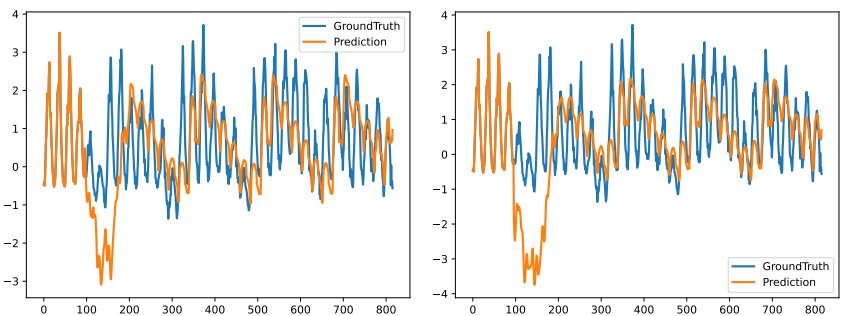

Figure 13: Univariate forecasting example of Autoformer on the ECL dataset with the prediction window size set to 720. Left figure shows the result of the original Autoformer (MSE: 0.80028, MAE: 0.66756). Right figure shows the result of the Autoformer trained with Lipschitz Regularizer ($\lambda = 5$, MSE: 0.91144, MAE: 0.69692).

**Multivariate Forecasting** We show four examples of multivariate forecasting in Figure 14, 15, 16, 17. Figure 14 and 16 show the cases where the Lipschitz Regularizer improves the performance of the model, while Figure 15 and 17 show two negative cases. For Figure 15, the sudden change in the input is abnormal, which influences the model with Lipschitz Regularizer more than the original

model. As for Figure 17, both models did not capture the pattern in the input data. This figure shows that Lipschitz Regularizer makes the output slightly more continuous, therefore causing the increased error.

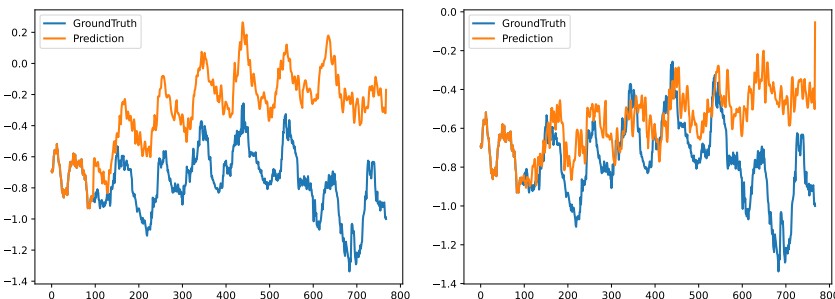

Figure 14: Multivariate forecasting example of Transformer on the ETTm1 dataset with the prediction window size set to 672. Left figure shows the result of the original Transformer (MSE: 1.17478, MAE: 0.83787). Right figure shows the result of the Transformer trained with Lipschitz Regularizer ($\lambda = 1$, MSE: 0.92590, MAE: 0.73367).

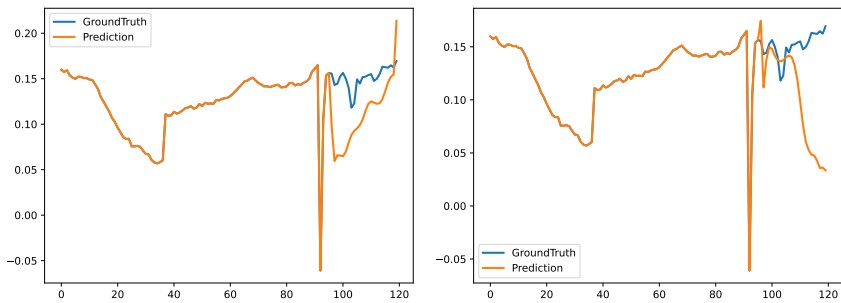

Figure 15: Multivariate forecasting example of Transformer on the Weather dataset with the prediction window size set to 24. Left figure shows the result of the original Transformer (MSE: 0.14897, MAE: 0.23293). Right figure shows the result of the Transformer trained with Lipschitz Regularizer ($\lambda = 1$, MSE: 0.21319, MAE: 0.28302).

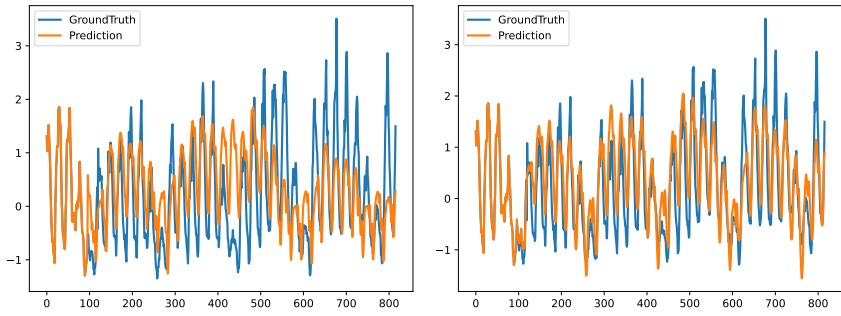

Figure 16: Multivariate forecasting example of Autoformer on the ECL dataset with the prediction window size set to 720. Left figure shows the result of the original Autoformer (MSE: 0.28641, MAE: 0.37610). Right figure shows the result of the Autoformer trained with Lipschitz Regularizer ($\lambda = 1$, MSE: 0.25691, MAE: 0.36375).

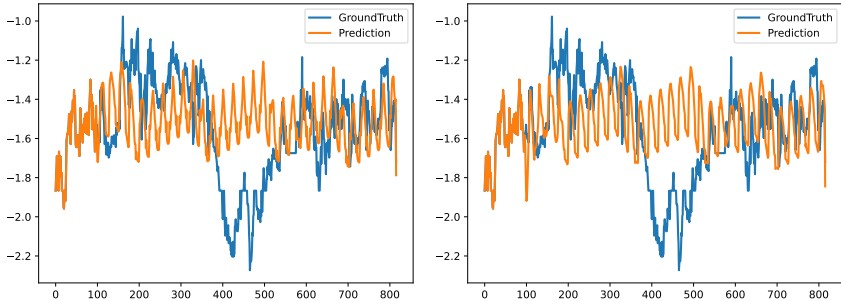

Figure 17: Multivariate forecasting example of Autoformer on the ETTh1 dataset with the prediction window size set to 720. Left figure shows the result of the original Autoformer (MSE: 0.47690, MAE: 0.49172). Right figure shows the result of the Autoformer trained with Lipschitz Regularizer ($\lambda = 1$, MSE: 0.50977, MAE: 0.50879).

## D    FINE-TUNE SWIN TRANSFORMER WITH LIPSCHITZ REGULARIZER

In this section, we investigate whether the proposed Lipschitz Regularizer can be used to improve large pre-trained models by fine-tuning its embedding layer on down-stream tasks. Here, we use a regular setting that the model is pre-trained on a large image dataset, and then fine-tuned it on the down-stream image classification task. Moreover, considering that Transformer-based models have shown great power in computer vision domains (Dosovitskiy et al., 2020; Liu et al., 2021), and these models process images by splitting them into patches and feeding the model a sequence of patch embeddings, we choose a typical model, i.e., Swin Transformer (Liu et al., 2021), for experiments in this section.

Generally, because input tokens of the Transformer are local image patches, they tend to be continuous, which might not match the preference of the Transformer model. Therefore, we apply the Lipschitz Regularizer to make them more discrete. Specifically, we use the Lipschitz Regularizer for outputs of the embedding layer, and change the loss as follows:

$$\mathcal{L}(y, \hat{y}, \hat{l}) = \mathcal{L}_{\text{Swin}}(y, \hat{y}) - \lambda \mathcal{L}_{\text{Lip}}(\hat{l}), \tag{10}$$

where $y$ is the ground-truth, and $\hat{y}$ is the output of the Swin Transformer. $\hat{l}$ is the output of the embedding layer, and $\mathcal{L}_{\text{Swin}}$ is the original loss of the Swin Transformer. $\lambda$ controls the magnitude of the Lipschitz Regularizer.

We use the pre-trained Swin Transformer on ImageNet[2], and fine-tune it on the image classification task with the Beans dataset (Lab, 2020), containing bean leaf images of diseased and healthy leaves. We only fine-tune the embedding layer and the last linear layer, and freeze other parts of the model.

Table 6: Results of fine-tuning Swin Transformer with Lipschitz Regularizer on an image classification task. Test accuracy with different $\lambda$ is reported.

| $\lambda$ | 5 | 1 | 0 | -1 | -5 |
|---|---|---|---|---|---|
| Test Accuracy | 0.9398 | **0.9549** | 0.9248 | 0.9098 | 0.8947 |

We show the validation accuracy during training in Figure 18 and list the testing accuracy in Table 6. By setting $\lambda$ to zero, we obtain the result of the baseline. We can see that the model performance can be improved by the Lipschitz Regularizer, showing great potential for changing data continuity in the fine-tuning setting and vision tasks. Besides, results also show that setting $\lambda$ to positive values (i.e., 5 and 1) benefits model performance, while setting it to negative values (i.e., -5 and -1) degenerates the performance. This verifies our intuition that the Transformer-based model prefers more discrete inputs.

---

[2]The pre-trained model is acquired at https://huggingface.co/microsoft/swin-base-patch4-window7-224

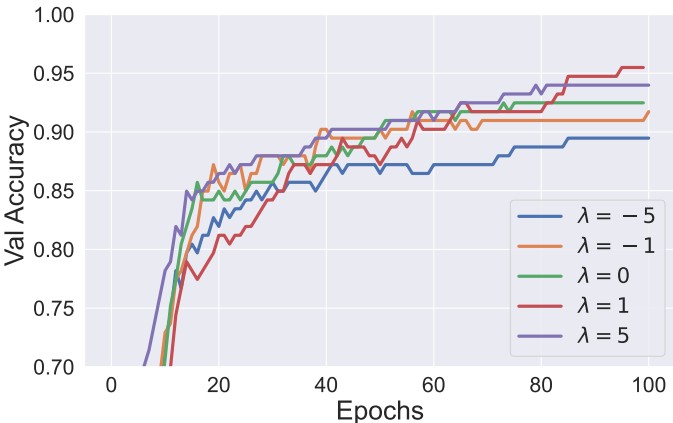

Figure 18: Validation accuracy of fine-tuning Swin Transformer in each epoch with different values of $\lambda$.

# E    APPLY LIPSCHITZ REGULARIZER TO THE SPEECH CLASSIFICATION TASK

In this section, we show the effectiveness of the Lipschitz Regularizer on the speech classification task by applying it to a Transformer-based model. As we discussed in § 1, the Transformer-based model prefers discrete inputs. However, voice signals are highly continuous since they are sampled from a continuous physical process with a high sample rate. This inspires us to use the Lipschitz Regularizer to make it more discrete and therefore more preferable for Transformer-based models.

Specifically, following the settings in Gu et al. (2021), we investigate the Performer model (Choromanski et al., 2020) on the Speech Commands (SC) dataset (Warden, 2018). We test the Performer model on two versions of SC. One is *MFCC*, where the sequence is pre-processed into standard MFCC features (length 161). Another is *Raw*, which contains unprocessed signals (length 16000). The Lipschitz Regularizer is applied after the embedding layer, and changes the loss as follows:

$$\mathcal{L}(y, \hat{y}, \hat{l}) = \mathcal{L}_{\mathrm{Per}}(y, \hat{y}) - \lambda \mathcal{L}_{\mathrm{Lip}}(\hat{l}), \tag{11}$$

where $y$ is the ground-truth, and $\hat{y}$ is the output of the Performer model. $\hat{l}$ is the output of the embedding layer, and $\mathcal{L}_{\mathrm{Per}}$ is the original loss of the Performer model. $\lambda$ controls the magnitude of the Lipschitz Regularizer.

Table 7: Results of the Performer model and Lipschitz Regularizer on the Speech Commands dataset. Test accuracy for MFCC and Raw speech data is reported.

| $\lambda$ | 0 | 1 | 3 |
|---|---|---|---|
| MFCC | 80.63 | 81.13 | **83.21** |
| Raw | 30.89 | **35.72** | 33.86 |

Results are shown in Table 7. The column with $\lambda = 0$ represents the baseline. The performance of the Performer model is increased by the Lipschitz Regularizer, which further verifies our claim that Transformer-based models prefer discrete inputs.

# F    NEURAL ODE WITH LIPSCHITZ REGULARIZER

In this section, we apply the Lipschitz Regularizer to the Neural ODE model (Chen et al., 2018) to see the effect of the regularizer on the model. Similar to the state-space model, Neural ODE is also a continuous-time model, which treats the input as samples from a continuous function. We expect that the Neural ODE will perform better when the input is more continuous.

We adopt the experiment of fitting time series using the latent ODE in its original paper (Chen et al., 2018). Essentially, the neural network is a generative latent function time-series model, predicting

the solution to an ODE, and the input data of this experiment is sampled from a randomly generated ODE with the same generation process as Chen et al. (2018). The network is a variational autoencoder, which consists of an RNN encoder and a Neural ODE decoder. To alter the continuity of the input to the Neural ODE, we directly apply Lipschitz Regularizer to the output of the RNN encoder as follows:

$$\mathcal{L}(y, \hat{y}, \hat{l}) = \mathcal{L}_{\text{ODE}}(y, \hat{y}) - \lambda \mathcal{L}_{\text{Lip}}(\hat{l}), \tag{12}$$

where $y$ is the ground-truth, and $\hat{y}$ is the output of the model. $\hat{l}$ is the output of the RNN encoder, and $\mathcal{L}_{\text{ODE}}$ is the original loss of the Neural ODE. $\lambda$ controls the magnitude of the Lipschitz Regularizer.

The MSE during training is shown in Figure 19. We can observe that Neural ODE performs better when data is more continuous. Predictions of 9 independent runs are presented in Figure 20. We can see that the model has better fitting results when we use Lipschitz Regularizer to make inputs more continuous.

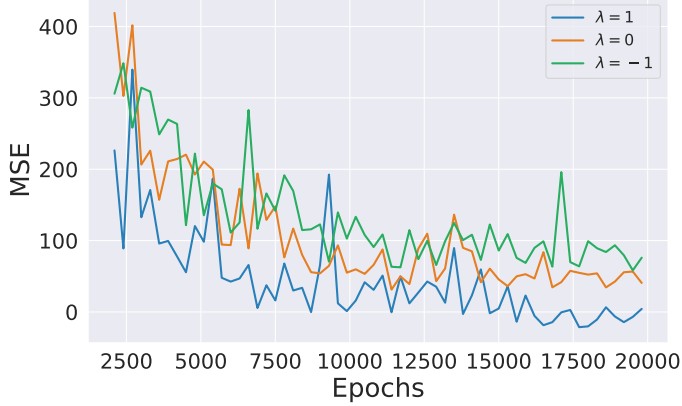

Figure 19: The MSE of the Neural ODE model with Lipschitz Regularizer during training.

## G  MATHEMATICAL DERIVATIONS

### G.1  DERIVATION OF EQUATION (4)

$$
\begin{aligned}
\sum_{i=0}^{n-1}(x_{i+1} - x_i)^2 &= \sum_{i=0}^{n-1}\left(\frac{f(t_{i+1}) - f(t_i)}{t_{i+1} - t_i}\right)^2 \\
&\approx \int_{\mathbb{R}}\left(\frac{\mathrm{d}f(t)}{\mathrm{d}t}\right)^2 \mathrm{d}t \\
&= \int_{\mathbb{R}} (2\pi i \xi)^2\, \hat{f}^2(\xi)(-\mathrm{d}\xi) \\
&= 4\pi^2 \int_{\mathbb{R}} \xi^2 \hat{f}^2(\xi)\mathrm{d}\xi \\
&= 4\pi^2 C \int_{\mathbb{R}} \xi^2 \frac{\hat{f}^2(\xi)}{C}\mathrm{d}\xi \\
&= 4\pi^2 C \mathbb{E}_{p(\xi)}[\xi^2]
\end{aligned} \tag{13}
$$

### G.2  CONTINUITY AND THE S4 MODEL

**Proposition G.1.** *Suppose $f_1, f_2 : \mathbb{R}_+ \to \mathbb{R}$ are two differentiable functions of input sequences, and their Lipschitz constant are $L_{f_1}$ and $L_{f_2}$. The HiPPO matrix with scaled Legendre measure (LegS) is denoted as HiPPO-LegS. Let the error of the HiPPO-LegS projection of $f_1, f_2$ at time $t$ be $\delta_1(t), \delta_2(t)$, respectively. Let $\hat{\delta}_1(t) = tL_{f_1}, \hat{\delta}_2(t) = tL_{f_2}$. For any time $t$, suppose $L_{f_1} \le L_{f_2}$, we have $\delta_1(t) = O(\hat{\delta}_1(t)), \delta_2(t) = O(\hat{\delta}_2(t))$, and $\hat{\delta}_1(t) \le \hat{\delta}_2(t)$.*

*Proof.* By Gu et al. (2020, Proposition 6), the LegS measure, which uniformly weighs all history, has the following property. Suppose the HiPPO-LegS projection for the target function $f(t)$ at time $t$ is $p^{(t)} = \text{proj}_t(f)$, then the error $\delta_f(t) = \|f_{\leq t} - p^{(t)}\| = O(tL_f/\sqrt{N})$, where $L_f$ is the Lipschitz constant of $f(t)$, and the maximum polynomial degree is $N - 1$. So, we have $\delta_1(t) = O(\hat{\delta}_1(t)), \delta_2(t) = O(\hat{\delta}_2(t))$, and $\hat{\delta}_1(t) \leq \hat{\delta}_2(t)$. Therefore, the error rate of HiPPO-LegS projection decreases with the Lipschitz constant, so with smaller Lipschitz constant, we expect smaller projection error. $\qquad\square$

### G.3 CONTINUITY AND THE RELU NETWORK

**Proposition G.2.** *Suppose there are two ReLU networks $g_{\theta_1}, g_{\theta_2}$ with identical architecture, and the Lipschitz constant of them are $L_1, L_2$, respectively. Let $h_1(\xi) = L_1/\|\xi\|^{n+1}, h_2(\xi) = L_2/\|\xi\|^{n+1}$, where $\xi$ is the frequency, and $\hat{g}_\theta(\xi)$ is the Fourier component of $g_\theta$. Suppose $L_1 \leq L_2$, we have $\hat{g}_{\theta_1}(\xi) = O(h_1(\xi)), \hat{g}_{\theta_2}(\xi) = O(h_2(\xi))$, and $h_1(\xi) \leq h_2(\xi)$.*

*Proof.* By Rahaman et al. (2019, Theorem 1), for a ReLU network $g_\theta$ with parameter $\theta$, its Fourier component is,

$$\hat{g}_\theta(\xi) = \sum_{n=0}^{d} \frac{G_n(\theta, \xi)}{\|\xi\|^{n+1}} \tag{14}$$

where the numerator $G_n(\theta, \cdot) : \mathbb{R}^d \to \mathbb{C}$ is bounded by $O(L_g)$. So, we have $\hat{g}_{\theta_1}(\xi) = O(h_1(\xi)), \hat{g}_{\theta_2}(\xi) = O(h_2(\xi))$, and $h_1(\xi) \leq h_2(\xi)$. Therefore, with smaller Lipschitz constant, we expect smaller $\hat{g}_\theta(\xi)$. $\qquad\square$

(a) Apply Lipschitz Regularizer with $\lambda = 1$.

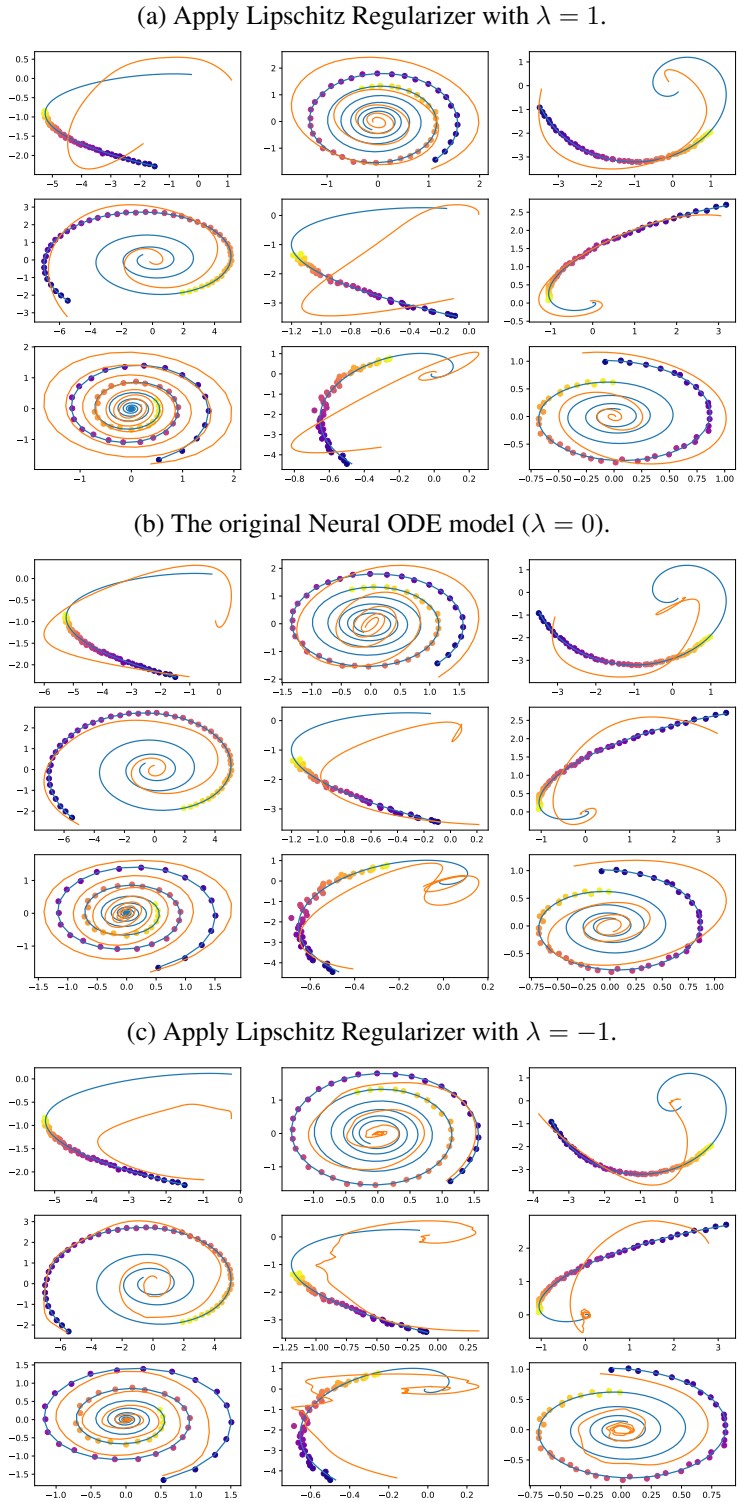

(b) The original Neural ODE model ($\lambda = 0$).

(c) Apply Lipschitz Regularizer with $\lambda = -1$.

Figure 20: Results of generative latent function with Neural ODE and Lipschitz Regularizer. Points are input sequences, and the color of points indicates their time. The blue line is the ground truth and the orange line is the prediction. The gap between points and the orange line indicates training loss, and the gap between the orange line and the blue line indicates the test loss.

