# OpenReview forum: "Data Continuity Matters: Improving Sequence Modeling with Lipschitz Regularizer"
_ICLR.cc/2023/Conference — ICLR 2023 notable top 25%_

### Official Review · Reviewer_L7Qs · 2022-10-24

**Confidence:** 3
**Correctness:** 4
**Technical Novelty And Significance:** 3
**Empirical Novelty And Significance:** 3
**Recommendation:** 8

**Clarity, Quality, Novelty And Reproducibility:**

The paper is clear and of high writing quality. The perspective of studying Lipschitz continuity property of sequential models is interesting and novel. I have no concerns with the reproducibility of the results.

**Strength And Weaknesses:**

Strength:

1. The main idea of the work is simple and clear. It is strongly motivated by empirical observations.
2. The analysis from both the time and frequency perspectives provides deeper insight and a solid basis for the proposed Lipschitz continuity regularizer.
3. The experiment results are coherent with the motivation and theoretical analysis.


Weakness:

The work only studied S4, informer, and ReLU networks. The theoretical results on the Lipschitz continuity property of S4 and ReLU network which in turn support the use of Lipschitz regularizer are model-specific. I’m concerned about the general applicability of the proposed Lipschitz regularizer to similar sequential models without more general theoretical or experiment results.


**Summary Of The Paper:**

The work studies the connection between sequence data continuity with the performance of deep sequential models and proposes a regularizer that adjusts the Lipschitz continuity of the model to improve its performance. It analyzes the property of the regularize from both time and frequency domains. Experiment results conform with the analysis and show the effectiveness of the proposed regularizer.

**Summary Of The Review:**

The perspective of studying the continuity property of sequential model is interesting and novel. The proposed Lipschitz regularizer is well supported by theoretical analysis and strong performance in experiment results. However, I'm concerned that the set of models studied in the paper, despite being exemplary, are limited.

---

> ### Author Response · Authors · 2022-11-18
> **Response to Reviewer L7Qs**
>
> Thank you for your time and effort in reviewing our paper. We respond to the weakness you mentioned in your review as follows.
>
> > The work only studied S4, informer, and ReLU networks. The theoretical results on the Lipschitz continuity property of S4 and ReLU network which in turn support the use of Lipschitz regularizer are model-specific. I’m concerned about the general applicability of the proposed Lipschitz regularizer to similar sequential models without more general theoretical or experiment results.
>
>
> We would like to clarify general and model-specific findings in this paper. Lipschitz regularizer measures the data continuity, which is the main focus of our paper, and is a general property of sequence data. Our finding that the model performance is highly related to the data continuity is also a general one. Then, considering that different models are proposed for different tasks, and they have diverse preferences on data continuity, we must turn to some model-specific investigations. However, the way to alter continuity for improving performance is applying the Lipschitz regularizer (with different coefficients), which is also the same across different models.
>
> Moreover, besides some typical models already involved (i.e., S4, Informer, and ReLU networks), we also add these experiments of more kinds of models in our revised version:
> - Two more Transformer-based models for time series forecasting in $\S$ 4.2 and Appendix C.2.
> - Fine-tune Swin Transformer in Appendix D.
> - Performer for speech classification in Appendix E.
> - Neural ODE as generative latent function time-series model in Appendix F.

---

> > ### Comment · Reviewer_L7Qs · 2022-12-08
> > **Update after Rebuttal**
> >
> > I'm satisfied with the authors response and I updated my score.

---

### Official Review · Reviewer_EzfA · 2022-10-25

**Confidence:** 2
**Clarity, Quality, Novelty And Reproducibility:** The paper has good novelty and clarit…
**Correctness:** 4
**Technical Novelty And Significance:** 3
**Empirical Novelty And Significance:** 3
**Recommendation:** 6

**Strength And Weaknesses:**

The paper has a good motivation, which is reasonable and straightforward;

The proposed regularizer is simple and general, can be used to improve performance of different models;

Good experimental results are obtained across different datasets on different models;

Some theoretical analysis and clear explanations are provided;

Some experiments can be added to strengthen the paper. Specifically, instead of training from scratch, I'm curious of whether we can use the proposed method to improve large pre-trained models by fine-tuning its embedding layer and freeze other layers on down-stream tasks, instead of fine-tuning the whole model or several selected layers. If the claims are correct, the proposed method should lead to good results. I think this experiment is very important, given the fact that many pre-trained transformer-like models are very important in real-world applications.

Is the finding in Figure 1 always holds across different dimensionalities?

**Summary Of The Paper:**

This paper analyzes the impact of data continuity on deep learning models, and propose Lipschitz regularizer which can improve performance of different models based on their preferred data continuity.

**Summary Of The Review:**

Based on  data continuity, the authors analyze the continuity preference of different models, and propose a simple yet effective regularizer to improve performance of different models. I lean towards acceptance of the paper.

---

> ### Author Response · Authors · 2022-11-18
> **Response to Reviewer EzfA**
>
> Thank you for your time and effort in reviewing our paper. We respond to the weaknesses and questions you mentioned in your review as follows.
>
> > Some experiments can be added to strengthen the paper. Specifically, instead of training from scratch, I'm curious of whether we can use the proposed method to improve large pre-trained models by fine-tuning its embedding layer and freeze other layers on down-stream tasks, instead of fine-tuning the whole model or several selected layers. If the claims are correct, the proposed method should lead to good results. I think this experiment is very important, given the fact that many pre-trained transformer-like models are very important in real-world applications.
>
> Thanks for this valuable suggestion! We have added this experiment in our revised version in Appendix D. Specifically, we use a regular setting that the model is pre-trained on a large image dataset (i.e., ImageNet), and then fine-tuned on the down-stream image classification task (i.e., Beans dataset [1]). We choose a typical and powerful Transformer-based model, i.e., Swin Transformer [2], and fine-tune its embedding layer with the Lipschitz regularizer. The loss function can be simply denoted as $L=L_\text{Swin}-\lambda L_\text{Lip}$, where $L_\text{Swin}$ is the original loss of the Swin Transformer, and $L_\text{Lip}$ is the Lipschitz regularizer. $\lambda$ controls the magnitude of the regularizer. We only fine-tune the embedding layer and the last linear layer, and freeze other parts of the model.
>
> | $\lambda$ | 5 | 1 | 0 | -1 | -5 |
> | - | - | - | - | - | - |
> | Test Accuracy | 0.9398 | **0.9549** | 0.9248 | 0.9098 | 0.8947 |
>
>
> Testing accuracy is listed in the above table. We can see that with positive $\lambda$, the model performance can be improved by the Lipschitz Regularizer, showing great potential for changing data continuity in the fine-tuning setting, also verifying our intuition that the Transformer-based model prefers more discrete inputs.
>
> [1] Makerere AI Lab. Bean disease dataset, January 2020. URL https://github.com/AI-Lab-Makerere/ibean/
>
> [2] Ze Liu, Yutong Lin, Yue Cao, Han Hu, Yixuan Wei, Zheng Zhang, Stephen Lin, and Baining Guo. Swin transformer: Hierarchical vision transformer using shifted windows. *In Proceedings of the IEEE/CVF International Conference on Computer Vision*, pp. 10012–10022, 2021.
>
>
> > Is the finding in Figure 1 always holds across different dimensionalities?
>
> Yes, it holds across different dimensionalities. The experiment in Introduction runs with univariant data, so we add a multivariate version of the experiment in Appendix A.2. Specifically, we generate 10 different input functions and map them to the output under the same setting, and we train the S4 model and Transformer model with the generated data. We show figures of 4 randomly selected dimensions in Appendix A.2, and the average MSE of all dimensions for each model is listed in the table below. Both figures and MSEs confirm that the finding holds across different dimensionalities.
>
> |  | High Continuity | Low Continuity |
> | - | - | - |
> | S4 | 0.0034 | 0.0909 |
> |S4 + Lip| - |0.0113|
> |Transformer| 0.0696| 0.0101|
> |Transformer + Lip| 0.0019| -|

---

### Official Review · Reviewer_cBYY · 2022-10-27

**Confidence:** 2
**Correctness:** 4
**Technical Novelty And Significance:** 3
**Empirical Novelty And Significance:** 3
**Recommendation:** 6

**Clarity, Quality, Novelty And Reproducibility:**

Paper is clearly written. Quality is good. Novelty cannot evaluate as I am not familiar with the type of data used.
Reproducibility: No link is provided to the code or the data.

**Strength And Weaknesses:**

Strengths:
- The methodology seems to be novel.
- Experimental evaluation looks extensive and quite thorough with different models and data types.

Weaknesses:
- The regulariser seems to be quite simple, it is just the square difference between two sequence values.
- Adding the regulariser did not always improve results but made them worse. I would have liked to see a detailed analysis on those situations, why that happens. E.g. For Pathfinder in table 1 and the quite a few cases (especially multivariate) in table 2.

**Summary Of The Paper:**

This paper proposes a Lipschitz regulariser to adjust to the continuity of certain type of data.

**Summary Of The Review:**

Paper seems interesting but I cannot evaluate its relevance to the field.

---

> ### Author Response · Authors · 2022-11-18
> **Response to Reviewer cBYY**
>
> Thank you for your time and effort in reviewing our paper. We respond to the weaknesses and questions you mentioned in your review as follows.
>
> > The regulariser seems to be quite simple, it is just the square difference between two sequence values.
>
> We indeed proposed a simple regularizer that only focuses on data continuity, but we also show that this simple regularizer is effective in sequence modeling, and analyzing the continuity preference for different models is non-trivial.
>
> Specifically, the square difference reflects the data continuity, which is a generic property existing in almost all sequence data. We show that the model performance is highly related to this property, and further changing the continuity by Lipschitz Regularizer can enhance the performance of multiple deep models. So, the importance and effecitiveness of the regularizer is verified. Moreover, different models prefer different data continuity, and the way to apply Lipschitz Regularizer is decided by the preference. In our paper, we take S4, Informer, and the ReLU network as case studies. They process data in totally different ways, and finding out their preference for data continuity is not easy.
>
> Generally, we believe that simple and effective methods are what we seek. Particularly, simplicity is the reason that this regularizer can be applied to many deep models without complex modifications.
>
> > Adding the regulariser did not always improve results but made them worse. I would have liked to see a detailed analysis on those situations, why that happens. E.g. For Pathfinder in table 1 and the quite a few cases (especially multivariate) in table 2.
>
> We have added some detailed analyses with figures for the S4 model on the Pathfinder dataset in Appendix B.2. We have also shown and explained some bad and good cases in the time series forecasting task in Appendix C.3.
>
> For the Pathfinder task, the performance drop is mainly caused by the embedding layer. As explained in $\S$ 4.1, since we cannot directly manipulate the underlying function of the input sequence, we add an extra embedding layer before the S4 layer. But the visualization for the output vector of the embedding layer shows that this embedding layer may overly and incorrectly blur or even erase some informative shapes in the original picture, causing some necessary information lost, and the model confused. For the time series forecasting task, we see a bad case with an abnormal sudden change in its original data, and another bad case happens when the base model fails to capture data pattern, and then Lipschitz Regularizer makes the output slightly more continuous, therefore causing the increased error.
>
> > Reproducibility: No link is provided to the code or the data.
>
> We have provided codes in Supplementary Material, and we will make it open source when the paper is accepted.

---

### Official Review · Reviewer_Tz2y · 2022-10-28

**Confidence:** 3
**Correctness:** 4
**Technical Novelty And Significance:** 4
**Empirical Novelty And Significance:** 4
**Recommendation:** 8

**Clarity, Quality, Novelty And Reproducibility:**

The overall quality of this paper is satisfactory. The comprehensive analysis and proposed method are novel. The reproducibility is high since the authors submit codes.

**Strength And Weaknesses:**

Strengths:

1) In my view, this paper is both comprehensive and deep. Now, there are many effective model structures such as Transformer and state space models for sequence modeling. But few works try to investigate the reasons why they work well in specific tasks. This paper starts from continuity, which is an intrinsic property of data, and clearly reveals the relationship between task data and model structures with theoretical supports. The authors provide meaningful insights into different tasks and model designs, which may guide the future research of sequence modeling.

2) In addition to analyzing the phenomenon, the authors also propose a simple and effective method by adjusting the data continuity according to model preferences. Empirical results show its superior performance on challenging tasks like LRA, which are convincing for me.

3) This paper is well-organized and easy to follow. I really enjoy reading this paper and learn a lot from it.

Weaknesses:

1) The authors only conduct experiments on LRA and time-series forecasting tasks in the analysis of Section 4. I wonder whether the authors try to use Lipschitz Regularizer in NLP or audio tasks such as language modeling because they are also attractive sequence modeling tasks.

2) Typo: Preformer –> Performer in line 7 of Section 2; Removing ) in line 1 of Section 5.1.


**Summary Of The Paper:**

This paper comprehensively investigates the continuity of sequential data from both theoretical and empirical perspectives. The authors observe that different model structures may prefer data with different continuity. Then, they give a theoretical analysis in both time and frequency domains. To further improve sequence modeling, they propose Lipschitz Regularizer which can flexibly adjusts data continuity based on the model preferences. Experimental results on different model structures and tasks demonstrate the effectiveness of Lipschitz Regularizer for many deep models in sequence modeling.

**Summary Of The Review:**

In my view, this paper is of high quality. I will surely recommend acceptance. But if other reviewers raise serious problems in this paper, I will be still open to change my mind.

---

> ### Author Response · Authors · 2022-11-18
> **Response to Reviewer Tz2y**
>
> Thank you for your time and effort in reviewing our paper. We respond to the weaknesses and questions you mentioned in your review as follows.
>
> > The authors only conduct experiments on LRA and time-series forecasting tasks in the analysis of Section 4. I wonder whether the authors try to use Lipschitz Regularizer in NLP or audio tasks such as language modeling because they are also attractive sequence tasks.
>
> Thanks for your great suggestion! We have added an experiment of applying the Lipschitz Regularizer to the speech classification task in Appendix E of the revised paper. Specifically, following the settings in [1], we investigate the Performer model [2] on two versions of the Speech Commands (SC) dataset [3]. One contains processed MFCC features and another contains raw signals. The Lipschitz Regularizer is applied after the embedding layer, and change the loss function as $L=L_\text{Per}-\lambda L_\text{Lip}$, where $L_\text{Per}$ is the original loss of the Performer model, and $L_\text{Lip}$ is the Lipschitz regularizer. $\lambda$ controls the magnitude of the regularizer. Since we think the Transformer-based model prefers discrete inputs, and voice signals are highly continuous since they are sampled from a continuous physical process with a high sample rate, we test some positive $\lambda$.
>
> | $\lambda$ | 0 | 1 | 3 |
> | - | - | - | - |
> | MFCC | 80.63 | 81.13 | **83.21**|
> | Raw | 30.89 | **35.72** | 33.86 |
>
> The results (shown in the table above) show that with positive $\lambda$, the model performance can be improved by the Lipschitz Regularizer, indicating that Transformer-based models prefer more discrete inputs.
>
> [1] Albert Gu, et al. Efficiently Modeling Long Sequences with Structured State Spaces. In *International Conference on Learning Representations*, 2021.
>
> [2] Krzysztof Marcin Choromanski, et al. Rethinking attention with performers. In *International Conference on Learning Representations*, 2020.
>
> [3] Pete Warden. Speech commands: A dataset for limited-vocabulary speech recognition. *arXiv preprint arXiv:1804.03209*, 2018
>
> > Typo: Preformer –> Performer in line 7 of Section 2; Removing ) in line 1 of Section 5.1.
>
> Thanks for pointing out! Fixed!

---

### Author Response · Authors · 2022-11-18
**General Response**

We greatly appreciate all reviewers for your time and effort in providing these insightful feedback that help us improve our work.

We are encouraged by the positive feedback on multiple aspects of our work: our paper is "both comprehensive and deep" ([Tz2y]) with reasonable and strong motivation ([EzfA, L7Qs]). Our method is "novel" ([cBYY]), "simple", and "effective" ([Tz2y, EzfA]). The experimental results are "convincing" ([Tz2y, EzfA]), "extensive and quite thorough" ([cBYY]), and "coherent with the motivation and theoretical analysis" ([L7Qs]). The theoretical analyses are "clear" ([EzfA]) and "provide deeper insight" into our method ([L7Qs]).

We provide point-to-point responses to each review in detail. We have submitted a revised version of the paper that highlights the changes in blue color. We summarize the changes as follows (the section numbers correspond to the revised version):

1. **[cBYY]** Experiments of two more Transformer-based models for time series forecasting are added in $\S$ 4.2 and Appendix C.2, with some examples for the time-series forecasting task in Appendix C.3.
2. **[EzfA]** The experiment in the introduction with multivariant data is added in Appendix A.2.
3. **[cBYY]** Analyses for the performance drop in the experiment of the S4 model on the Pathfinder dataset are added in Appendix B.2.
4. **[EzfA]** A new experiment about fine-tuning Swin Transformer is added in Appendix D.
5. **[Tz2y]** A new experiment about the Performer for speech classification is added in Appendix E.
6. A new experiment about Neural ODE as the generative latent function time-series model is added in Appendix F.
7. Due to the page limitation, we have moved mathematical derivations to Appendix G to provide space for new experiments and their analyses.

---

### Decision · Program_Chairs · 2023-01-20

**Decision:**

Accept: notable-top-25%

**Justification For Why Not Higher Score:**

The datasets and models studied are somewhat limited.

**Justification For Why Not Lower Score:**

Quoting: This work proposes a simple regularization method to better adapt model architecture to continuity properties of time series data. They find that this regularizer can usefully adjust architectural continuity priors to better fit the data they are being trained and evaluated on, leading to benefits on a variety of tasks and will make a nice addition to ICLR.

**Metareview: Summary, Strengths And Weaknesses:**

This work proposes a simple regularization method to better adapt model architecture to continuity properties of time series data. They find that this regularizer can usefully adjust architectural continuity priors to better fit the data they are being trained and evaluated on, leading to benefits on a variety of tasks and will make a nice addition to ICLR.

Nit: Figure 1 caption first raw -> first row

**Note From Pc:**

if the above contains the word "oral" or "spotlight" please see: "oral" presentation means -> notable-top-5% and "spotlight" means -> notable-top-25%. As stated in our emails, we are disassociating presentation type from AC recommendations